# Evolution of tropospheric aerosols over central China during 2010-2024 as observed by lidar

Dongzhe Jing<sup>1,2,3</sup>, Yun He<sup>1,2,3,\*</sup>, Zhenping Yin<sup>4</sup>, Kaiming Huang<sup>1,2,3</sup>, Fuchao Liu<sup>1,2,3</sup>, Fan Yi<sup>1,2,3</sup>


Abstract. Air quality in China has improved significantly over the past decade. However, recent studies show that this progress has notably slowed in recent years. To investigate regional patterns and driving factors, we examined the long-term evolution of tropospheric aerosols over Wuhan (30.5°N, 114.4°E) from 2010 to 2024, using ground-based polarization lidar observations. Aerosol optical depth (AOD) trends are divided into two phases: a declining trend (-0.077 yr¹) during 2010-2017 (Stage I) and a fluctuating period during 2018-2024 (Stage II). Contributions from natural (dust) and anthropogenic (non-dust) aerosols were analyzed separately. Dust optical depth (DOD) consistently declined (-0.011 yr¹) until August 2020 and became larger again afterwards. In Stage I, anthropogenic aerosols (-0.068 yr¹) were responsible for 88.3% of the total AOD reduction, primarily due to decreases in boundary-layer AOD. In Stage II, anthropogenic AOD fluctuated, possibly due to atmospheric chemistry and meteorological factors. Seasonal variations were also observed. Anthropogenic aerosols appeared from the surface to 2.5 km in summer, with particle extinction and mass concentration of 0.12 km⁻¹ and 83.0 μg m⁻³, which were concentrated below 0.7 km in winter, with much higher particle extinction and mass concentration of 0.31 km⁻¹ and 211.8 μg m⁻³. Two case studies highlighted typical pollution events: summertime transboundary agricultural biomass burning smoke in June 2014 and wintertime local anthropogenic aerosol pollution in January 2019. These findings improve our understanding of how regional aerosols respond to local emission controls and long-range transport of dust and smoke.

<sup>&</sup>lt;sup>1</sup>School of Earth and Space Science and Technology, Wuhan University, Wuhan 430072, China

<sup>5 &</sup>lt;sup>2</sup>Key Laboratory of Geospace Environment and Geodesy, Ministry of Education, Wuhan 430072, China.

<sup>&</sup>lt;sup>3</sup>State Observatory for Atmospheric Remote Sensing, Wuhan 430072, China.

<sup>&</sup>lt;sup>4</sup>School of Remote Sensing and Information Engineering, Wuhan University, Wuhan 430072, China.

<sup>\*</sup>Correspondence to: Yun He (heyun@whu.edu.cn)

#### 1 Introduction






Atmospheric aerosols play an essential role in the global climate by affecting the radiation budget through the absorption or scattering of solar radiation (Huang et al., 2014; Bellouin et al., 2020; Liu and Matsui, 2021). In addition, aerosols act as cloud condensation nuclei or ice nucleating particles, significantly influencing cloud properties through aerosol-cloud interactions (ACI) (Rosenfeld et al., 2014; Kanji et al., 2017; Yin et al., 2021a; He et al., 2021, 2022a). ACI processes currently contribute the largest uncertainty in global effective radiation forcing (IPCC, 2021). Furthermore, aerosols are closely linked to the living environment and human health. Particulate matter with a diameter <2.5 μm (PM<sub>2.5</sub>) is one of the most important air pollutants, capable of being inhaled into the lungs, thereby degrading air quality, reducing visibility, and damaging human health (van Donkelaar et al., 2015; Cohen et al., 2017).

During the first decade of the 21st century, rapid urbanization and industrialization in China caused severe air pollution, leading to a significant increase in PM<sub>2.5</sub> concentrations and aerosol optical depth (AOD) (Xie et al., 2016; Hu et al., 2021a; de Leeuw et al., 2022). To address this issue, the Chinese government implemented the "*Technical guidelines for air pollution control projects*" (Ministry of Ecology and Environment of the People's Republic of China, 2011), "*Air Pollution Prevention and Control Action Plan*", "*Three-year Plan on Defending the Blue Sky*", and "*Action Plan for continuous improvement of air quality*" to improve air quality (The State Council of the People's Republic of China, 2013, 2018, 2023; Geng et al., 2019). As a result, SO<sub>2</sub> and NO<sub>3</sub> concentrations decreased by 59% and 21%, respectively, between 2013-2017 (Zheng et al., 2018). The weighted annual mean PM<sub>2.5</sub> concentrations also declined from 61.8 to 42.0 μg m<sup>-3</sup> during this period (Zhang et al., 2019). Similarly, Papachristopoulou et al. (2022) reported a significant negative AOD trend for the Chinese megacities with up to -0.3 per decade during 2003-2020. However, a recent study found that the declining rate of PM<sub>2.5</sub> concentration has slowed down, from -4.5 μg m<sup>-3</sup> yr<sup>-1</sup> in 2013-2017 to -2.3 μg m<sup>-3</sup> yr<sup>-1</sup> after 2018; in contrast, the average cost of reducing one unit of PM<sub>2.5</sub> concentration has risen significantly from 49 billion RMB per μg m<sup>-3</sup> from 2013-2017 to 100 billion RMB per μg m<sup>-3</sup> after 2018 (Geng et al., 2024), indicating the weakened emission control measures.

In our previous study, we reported a consistent downward trend in AOD during 2010–2020 (Yin et al., 2021b). However, it appears that this decline in AOD ceased and even slightly reversed after 2018 (Yin et al., 2021). Given that the reduction in surface PM<sub>2.5</sub> concentrations also slowed after 2018 (Geng et al., 2024), it remains unclear whether the consistent downward trend in AOD merely decelerated or completely halted at some point after 2018. In addition, in the additional four years following 2020, several factors including industrial shutdowns during the COVID-19 pandemic, abnormal Asian dust events (Gui et al., 2022; He et al., 2022b), and extreme precipitation (Wang et al., 2023a; Li et al., 2024) may have further influenced AOD levels in Wuhan. Therefore, it is of great interest to extend the analysis by incorporating more recent datasets from our polarization lidar observations to assess the impact of the weakened emission control measures on AOD over Wuhan. In particular, polarization lidar can distinguish particle backscatter from dust and non-dust aerosol components, making it valuable for analyzing the evolution of natural and anthropogenic aerosol sources (Zhang et al., 2024). This extended analysis would contribute to evaluating the effectiveness of the Chinese government's aerosol emission control policies.

In addition, the vertical distribution of aerosols plays an important role in both global climate and human living environment. The interaction between aerosols and the boundary layer (BL) significantly affects atmospheric stability, while aerosols with extended residence times in the free troposphere (FT) can have more prolonged effects on the climate (Bourgeois et al., 2018).

During some air pollution episodes, long-range transported agricultural biomass burning smoke (ABBS) and desert dust are frequently observed in the FT, subsequently interacting with the BL when they deposit (Hänel et al., 2012; Chen et al., 2014; Chen et al., 2017; Lolli et al., 2023). Benefiting from high spatiotemporal resolution, lidar is the most favorable approach for observing aerosol vertical distribution. For instance, during a mega Asian dust event in March 2021, an elevated dust plume was initially detected by ground-based lidar over Wuhan nearly a day before the ground PM<sub>10</sub> concentration began to rise (He et al., 2022b). Since October 2010, we have conducted routine and consistent lidar observations on the campus of Wuhan University in Wuhan (Yin et al., 2021b; He et al., 2024; Jing et al., 2024), a mega industrial city in central China influenced by local anthropogenic aerosol emissions as well as long-range transported dust and smoke plumes (Lu et al., 2018; Jing et al., 2023). This study serves as a valuable supplement to the long-term evaluation of the optical properties of tropospheric aerosols in China.

This study analyzes the long-term characteristics of tropospheric aerosols over Wuhan from October 2010 to September 2024, based on ground-based polarization lidar observations, together with satellite data, reanalysis data, and surface environment monitoring data. The paper is organized as follows. Section 2 provides a brief description of the instruments and data processing methods employed. Section 3 presents the statistical characteristics of tropospheric aerosols over Wuhan. In Section 4, two typical air pollution events are discussed: one caused by summertime ABBS intrusion and the other by wintertime haze pollution in the low troposphere. In the last section, a summary and conclusions are presented.

## 2 Instrumentation, data, and methodology

## 2.1 Polarization lidar in Wuhan



A 532-nm polarization lidar has been routinely operated to observe the vertically resolved optical properties of aerosols over Wuhan (30.5°N, 114.4°E) since October 2010 (Kong and Yi, 2015). Raw lidar data were stored with resolutions of 1 minute and 30 meters. The lowermost height with complete field-of-view (FOV) observation is 0.3 km. The volume (aerosol + molecular) depolarization ratio  $\delta_v$  (VDR) is calculated as the ratio of perpendicular- to parallel-oriented signals, which can be used to derive the particle depolarization ratio  $\delta_p$  (PDR) (Freudenthaler et al., 2009). PDR values reflect the non-spherical characteristics of particle shapes, which are in general 0-0.075 for polluted continental/smoke aerosols, 0.075-0.2 for polluted dust, and >0.2 for pure dust (Kim et al., 2018). Note that hygroscopic growth can alter particle shapes, resulting in a decrease in PDR. Fernald method was used to retrieve the total particle backscatter coefficient  $\beta_p$  and particle extinction coefficient  $\alpha_p$ , assuming a fixed lidar ratio of 50 sr for the entire statistics, 70 sr for the smoke case, and 57 sr for the urban/industrial aerosol case (Haarig et al., 2018; Zhang et al., 2021a). Moreover, the dust and non-dust backscatter (or extinction) coefficient

can be calculated by the polarization-lidar photometer networking (POLIPHON) method (Tesche et al., 2009). The mass concentration for non-dust component ( $M_{\rm nd}$ ), which is considered anthropogenic aerosols in this study, are calculated from particle extinction, by multiplying by particle density (1.55 g cm<sup>-3</sup>) and extinction-to-volume conversion factor (0.44×10<sup>-12</sup> Mm) (He et al., 2022b). Aerosol optical depth (AOD) is defined as the integral of the particle extinction coefficient for a certain altitude range. In this study, the tropospheric AOD and dust optical depth (DOD) can be calculated by:

$$AOD = \int_{Z_{b}}^{Z_{t}} \alpha_{p}(z) dz$$
 (1)

$$DOD = \int_{z_b}^{z_t} \alpha_d(z) dz$$
 (2)

where  $Z_b$  and  $Z_t$  are the lower (base) and upper (top) limits of the integration height, respectively. The  $Z_b$  was set to 0 km, extinction coefficient below 0.35 km were assumed equal to that at 0.35 km, possibly causing an uncertainty of <0.05 in AOD (Baars et al., 2017). The  $Z_t$  was set to 7 km to ensure a sufficient signal-to-noise ratio (Yin et al., 2021b). The non-dust AOD was derived by subtracting DOD from AOD. The uncertainties in the lidar-derived parameters are listed in Table 1. The uncertainty in non-dust extinction is relatively larger due to variability in lidar ratio for continental aerosols, which can be 35 sr for non-absorbing anthropogenic haze and 75 sr for absorbing biomass-burning smoke (Mamouri and Ansmann, 2016).

Table 1. Estimated uncertainties in the lidar-derived parameters.



| Parameter                                        | Uncertainty | Reference                        |
|--------------------------------------------------|-------------|----------------------------------|
| Volume depolarization ratio $\delta_{ m v}$      | <5%         | Kong and Yi (2015)               |
| Particle depolarization ratio $\delta_{\rm p}$   | 5-10%       | Mamouri et al. (2013)            |
| Particle backscatter coefficient $\beta_p$       | <10%        | Zhuang and Yi (2016)             |
| Particle extinction coefficient $\alpha_p$       | <20%        | Kafle and Coulter (2013)         |
| Dust backscatter coefficient $\beta_d$           | 10-30%      |                                  |
| Dust extinction coefficient $\alpha_d$           | 15-25%      |                                  |
| Non-dust backscatter coefficient $\beta_{ m nd}$ | 10-30%      | Mamouri and Ansmann (2016, 2017) |
| Non-dust extinction coefficient $\alpha_{nd}$    | 20-40%      |                                  |
| Non-dust mass concentration $M_{\rm nd}$         | 30-40%      |                                  |

After cloud screening, cloud-free profiles with a duration time of 30-80 minutes and a vertical resolution of 30 meters are obtained (Yin et al., 2021b). From October 2010 to September 2024, in total of 24910 cloud-free profiles were derived from 2139 days for further statistical study (Figure 1). Since the installation of a transparent waterproof window in 2017, the lidar system can continuously perform regardless of rainy or snowy conditions, except for occasional maintenance (Yi et al., 2021). Therefore, the number of observation days can even exceed 300 days in recent years (2021-2023).

To improve the representativeness of the long-term statistics, several data quality control protocols have been applied in our calculations. Aerosol hygroscopic growth would significantly enhance the observed particle extinction in high humidity conditions (Zieger et al., 2013), which is several times larger than the dry particle extinction. During severe haze episodes, the particle extinction over Wuhan would not exceed 1.5 km<sup>-1</sup> according to our previous findings (Zhang et al., 2021a). A cloud-free profile having at least one vertical bin with  $\alpha_p>1.5$  km<sup>-1</sup> was removed to avoid the influence of fog. In total, 676 cloud-free profiles, accounting for 2.7% of the total number of profiles, were removed. Additionally, for representativeness, monthly mean AODs and extinction coefficients were considered effective only if the number of cloud-free profiles exceeded 15.

Figure 1. Annual count of days with lidar observations and valid retrievals of aerosol optical parameter profiles (October 2010 to September 2024).

# 2.2 Satellite data



In addition to ground-based lidar detection, observational data from spaceborne instruments were utilized to depict the horizontal distribution of the aerosol plumes. The Cloud-Aerosol Lidar with Orthogonal Polarization (CALIOP) on board the Cloud-Aerosol Lidar and Infrared Pathfinder Satellite Observation (CALIPSO) launched in 2006 aims to observe the vertically optical properties of cloud and aerosol (Winker et al., 2010). In this study, CALIOP version 4.51 level-1B and level-2 data products were used to observe the vertical/horizontal distribution and optical properties of a biomass-burning smoke event in June 2014. The elastic backscatter, extinction, and depolarization ratio at 532 nm are measured with vertical and horizontal

resolutions of 30 m and 1/3 km below an altitude of 8.2 km for level-1B data, and 60 m and 5 km below 20.0 km for level-2 data.

The Ozone Mapping and Profiler Suite (OMPS), carried on the Suomi National Polar-orbiting Partnership (Suomi NPP) satellite launched in October 2011 (Jaross et al., 2014), comprises a downward-looking nadir mapper, a nadir profiler, and a limb profiler. Among them, the nadir mapper provides the measurement of UV Aerosol Index (UVAI) with a spatial resolution of 50×50 km, which is an effective indicator of elevated UV-absorbing aerosols (e.g., smoke and dust) (Lee et al., 2015; Tao et al., 2022). In this study, the UVAI provided by OMPS-NPP level 2 data was used to exhibit the spatial evolution of the biomass-burning smoke event in June 2014 near Wuhan.

The Moderate Resolution Imaging Spectroradiometer (MODIS) instrument carried on the Terra satellite continually collects data in 36 spectral channels with global coverage every 1-2 days since December 1999. The daily corrected reflectance is obtained by removing gross atmospheric effects, such as Rayleigh scattering, from MODIS visible bands 1-7. The productions are provided as near-real-time imagery with a spatial resolution of 0.25 km, which can be used for monitoring wildfires and smoke plumes (Gumley et al., 2010). Moreover, the fire and thermal anomalies are primarily derived from MODIS micrometer radiances, which would be absorbed and scattered by clouds. In this study, the corrected reflectance imageries with the fire and thermal anomalies provided by MODIS on Terra were used to show the source of the biomass-burning smoke event in June 2014.

## 140 2.3 HYSPLIT model






The NOAA/ARL (National Oceanic and Atmospheric Administration/Air Resources Laboratory) Hybrid Single Particle Lagrangian Integrated Trajectory (HYSPLIT) model can simulate the forward and backward trajectories of air mass by initializing the starting time, altitude, and geographical location (Draxler and Rolph, 2003; Stein et al., 2015). The meteorological field data from the GDAS archive (Kanamitsu, 1989) were used to drive the model. In this study, the simulated backward trajectories were used to track the transport pathway of the biomass-burning smoke event in June 2014.

## 2.4 Air pollution monitoring data

A nationwide air quality monitoring network was established by the Chinese National Environmental Monitoring Center (CNEMC) in 2013, providing an hourly dataset of PM<sub>2.5</sub>, PM<sub>10</sub>, SO<sub>2</sub>, NO<sub>2</sub>, CO, and O<sub>3</sub> afterward (Tong et al., 2022). Previous studies show that biomass-burning aerosols and local industrial emissions can strongly increase PM<sub>2.5</sub>, CO, SO<sub>2</sub>, and NO<sub>2</sub> concentrations (Xu et al., 2019; Zhang et al., 2021a). In this study, the PM<sub>2.5</sub>, NO<sub>2</sub>, SO<sub>2</sub>, and CO concentrations were used to study the long-term variation during 2014-2024 as well as the surface environment in two severe pollution events. The PM<sub>2.5</sub> concentration will be used to assess the contribution of human emission factors to the variations in non-dust AOD.

## 2.5 ERA5 reanalysis data




The European Center for Medium-Range Weather Forecasts (ECMWF) reanalysis version 5 (ERA5) is atmospheric reanalysis data of the global climate covering the period from January 1940 onwards, providing the hourly estimates of atmospheric, land, and oceanic climate variables (Hersbach et al., 2020). The boundary layer (BL) height from ERA5 is derived from the ECMWF Integrated Forecasting System's turbulent diffusion and turbulent orographic form drag schemes (Hersbach et al., 2024). In this study, the ERA5 hourly BL height data at 30.5°N and 114.4°E were used to separate non-dust AOD from the BL and FT.

In addition, the ERA5 hourly relative humidity (%) and u-/v- component wind speed (WS, m s<sup>-1</sup>) within 1000-850 hPa, total precipitation (TP, m), vertical velocity (VV, Pa s<sup>-1</sup>) at 850 hPa, and boundary layer height (BLH, m) were used to estimate the contribution of meteorological factors to anthropogenic emissions, which were estimated as the largest meteorological contributions for AOD variation in south China during 1998-2014 (Che et al., 2019). Among them, the monthly RH, WS, VV, and BLH were calculated by hourly data corresponding to the time of the lidar profile. However, due to our lidar profiles are cloud-free, the monthly precipitation was determined by selecting the precipitation data on the day when there were lidar observations. The VV represents the speed of air motion in the upward or downward direction. Due to the pressure-based vertical coordinate system used by the ECMWF Integrated Forecasting System, the negative values of vertical velocity indicate upward motion.

## 2.6 Methodology of the Lindeman, Merenda, and Gold (LMG)

The LMG method was used to quantify the relative contributions of meteorological or human emission factors on the variation of anthropogenic AOD. For the number of p variations as inputs, it first calculates all possible permutations of variable addition (p! different arrangements). Then calculates the sequential  $R^2$  increment when the variable is added for each sequence, that is, the additional explanatory power brought by this variable. Finally, calculating the average of the increments for all permutations, and the average contribution of each variation can be obtained. The LMG method for the ith variation  $x_i$  can be expressed as (Che et al., 2019):

$$LMG(x_i) = \frac{1}{p!} \sum_{\substack{r \text{ permutation}}} \operatorname{seq} R^2(\{x_i\}|r)$$
(3)

where r represents the rth permutation as r=1, 2, ..., p!. The seq  $R^2(\{x_i\}|r)$  represents the sequential sum of  $R^2$  in the ordering of the regressors in the rth permutation. In this study, the LMG method derived relative contributions for meteorological or human emission factors on the variation of anthropogenic AOD were provided.

# 3.1 Overview of aerosol optical properties

Figure 2a shows the vertical distribution of the particle extinction coefficient over Wuhan from October 2010 to September 2024. Tropospheric aerosols are mainly concentrated within the boundary layer below 2.0 km, consistent with CALIOP observations in central China (109-116°E and 26-33°N) reported by Lu et al. (2018). Below 1.0 km, severe air pollution with  $\alpha_p$  values exceeding 0.6 km<sup>-1</sup> were more frequent before 2015, suggesting a significant improvement in air quality in recent years. In Figure 2b, two stages can be preliminarily distinguished: an evident declining phase as Stage I and a fluctuating phase as Stage II, with the transition occurring around 2017 and 2018. To determine the transition point between these two stages, we tested different scenarios by shifting the transitional month (i.e., the end of Stage I) from January 2017 to December 2018. By balancing the steepest decline in Stage I and a slope that is closest to zero in Stage II, the transition most likely appeared within the first 2-3 months of 2018. For simplicity, we adopted December 2017 as the end of Stage I and January 2018 as the beginning of Stage II. The monthly average AODs show a declining trend with a rate of -0.077 yr<sup>-1</sup> during 2010-2017 (Stage I), and fluctuated during 2018-2024 (Stage II). Using MODIS Multi-Angle Implementation of Atmospheric Correction (MAIAC) AOD data, de Leeuw et al. (2022) also found that annual mean AOD stabilized with only a 10% fluctuation around the average across China after 2017, contrasting with the significant decrease in AOD during 2011-2017. Moreover, annual mean AOD over Shanghai (31.0°N, 121.5°E) and Zhengzhou (34.5°N, 113.5°E) also ceased to decline and stabilized at values of 0.4-0.5 after 2018 (de Leeuw et al., 2021).

The significant reduction in AOD during Stage I (2010-2017) is mainly attributed to the decrease in three components: local anthropogenic aerosols, regional transboundary smoke, and long-range transported dust. First, the strict policies of anthropogenic emissions control, in particular China's Clean Air Action implemented in 2013, resulted in a substantial reduction in local anthropogenic aerosol emissions. Zhang et al. (2019) reported that the national population-weighted annual mean PM<sub>2.5</sub> concentrations decreased from 61.8 to 42.0 μg m<sup>-3</sup> between 2013-2017, attributed to strengthened industrial emission standards, the upgrading of industrial boilers, the phasing out of outdated industrial capacities, and the promotion of clean fuels. Second, the widespread enforcement of straw-burning bans during 2013-2018 reduced the long-range transported smoke particles that intruded into Wuhan (Huang et al., 2021). During the harvest season in summer/autumn, large amounts of agricultural biomass burning smoke (ABBS) were generated by the centralized burning of agricultural straws in a short period, resulting in extremely large monthly mean AOD, such as 1.43 in June 2012, and 1.18 in June 2014 (2.7 and 2.3 times compared to grand mean AOD in 2010-2024 with 0.52, in Figure 2b). Since the implementation of straw-burning bans, no extreme events with AOD >0.8 have been observed after 2015. Huang et al. (2021) also found a 46.9% reduction in the national total PM<sub>2.5</sub> originating from ABBS during 2013-2018. Third, a decrease in surface wind speed and an increase in vegetation cover over northwestern China led to reduced dust activities in major dust sources, i.e., the Gobi Desert (GD) and Taklimakan Desert (TD), which in turn decreased the dust intrusion into the downstream regions (An et al., 2018). Our previous study also

confirmed that the dust optical depth (DOD) in Wuhan showed a decrease rate of -0.011 yr<sup>-1</sup> from 2010 to 2020 (Jing et al., 2024).

Figure 2. (a) Time-altitude contour plots of the monthly mean particle extinction coefficient derived from 532-nm polarization lidar observations over Wuhan from 2010 to 2024. White stripes indicate periods with data unavailable due to weather conditions, hardware maintenance, or fewer than 15 cloud-free profiles being recorded in a given month. (b) AOD values for the monthly mean (blue dots) and individual cloud-free profile (light blue dots) over Wuhan during 2010-2024. The orange solid and dashed lines represent the linear fits of the monthly mean AOD for the periods 2010-2017 and 2018-2024, respectively. A box plot is presented for each year during 2011-2024, with the center line indicating the median value and the bottom and top edges representing the 25th and 75th percentiles, respectively. The whiskers are set to be 1.5, and hollow circles denote outliers. Gray solid dots represent the annual mean values.

The situation for Stage II (2018-2024) differs significantly from Stage I. In Stage II, AOD generally shows a relatively fluctuating trend. This is because the effectiveness of China's Clean Air Action has decreased after the initial Stage of rapid improvement. Geng et al., (2024) found that the average cost of reducing one unit of PM<sub>2.5</sub> concentration after 2018 (100 billion RMB per μg m<sup>-3</sup>) was twice that before 2018 (49 billion RMB per μg m<sup>-3</sup>), resulting in a slower decreasing rate of PM<sub>2.5</sub> concentration (-2.3 μg m<sup>-3</sup> yr<sup>-1</sup> after 2018 versus -4.5 μg m<sup>-3</sup> yr<sup>-1</sup> during 2013-2017). Furthermore, aerosol chemistry, i.e., the balance between different pollutants, is another key factor. For example, a decrease in sulfate emissions weakened the ability to neutralize ammonia (NH<sub>3</sub>) (Liu et al., 2018), leading to excess NH<sub>3</sub> that forms particulate nitrate, which in turn reduces the effectiveness of control policies (Geng et al., 2024). This may lead to severe air pollution in January 2019 (with a

225

monthly mean AOD of 0.6). In addition, the fluctuations in AOD during Stage II can also be attributed to meteorological factors. The annual mean AOD increased from 0.37 to 0.49 during 2018-2019. The precipitation near Wuhan in 2019 decreased by approximately 20-50% compared to the average value of previous years (Zeng et al., 2020), reducing the wet scavenging effect of aerosols (Li et al., 2023). During the winter of 2020-2021, AOD values were relatively low (~0.2-0.3), due to the 'Warm Arctic-Cold Siberia' pattern, which caused cold air from high latitudes to move southward to East China (Zhang et al., 2021b). This led to strong winds and cold waves, aiding the cleaning of the air. After 2021, the increased monthly mean AOD of 0.46-0.66 in spring was partly due to enhanced dust intrusion from north and northwest China, influenced by an exceptionally strong Mongolian cyclone and surface meteorological anomalies (Gui et al., 2022; Chen et al., 2024).

#### 3.2 Dust variation trend

Dust aerosols mainly originate from major desert regions in East Asia, i.e., the GD and TD, and generally intrude into Wuhan during spring (DOD: 0.21) and winter (DOD: 0.15). In summer, however, the prevailing southeastern monsoon suppresses the southeastward transport of dust plumes, leading to extremely rare dust intrusions, with a DOD of 0.02, i.e., an order of magnitude lower than in spring (Jing et al., 2024). Note that we cannot completely rule out the potential contribution of anthropogenic dust to the lidar-derived DOD. In a megacity such as Wuhan, direct anthropogenic dust is likely more important than indirect anthropogenic dust emitted from non-urbanized surfaces, e.g., cropland, pastureland, and dry lakes. However, Chen et al. (2019) reported that the major direct anthropogenic dust emissions in China are concentrated in Northern China, while emissions over central China are significantly lower. Therefore, it is reasonable to assume that the lidar-derived DOD primarily reflects contributions from desert dust (i.e., natural sources). Dust particles are generally blown up by surface wind in the deserts, and then transported southward by cold air and eastward via prevailing westerly. Figure 3a shows the evolution of DOD over Wuhan from 2010 to 2024. A clear downward trend in DOD can be observed in the early years. Unlike the total AOD, which ceased decreasing from 2018, our analysis reveals that the DOD decline continued until the end of 2020, with a rate of -0.011 yr<sup>-1</sup> during the period of 2010-2020, primarily driven by meteorological control over dust emissions. The decrease in wind speed and the increase in vegetation cover in northwest China together facilitated the reduction in natural dust emissions prior to 2020 (Xu et al., 2006; An et al., 2018).

Figure 3. Monthly mean (a) DOD and (b) the ratio of DOD to total AOD from 2010 to 2024. A box plot is presented for each year during 2011-2024, with the gray solid dots representing the annual mean values. The dark orange line represents the linear fit of the monthly mean DOD from 2010 to 2020. The parameters for box plots are the same as in Figure 2.

However, extreme springtime dust events have occurred again in recent years, with average DOD values ranging from 0.16 to 0.20 in spring during 2021-2024, which are larger than the spring mean DOD of 0.12 in 2020 (the last year of the downward trend). In March 2021, an exceptionally strong Mongolian cyclone, along with a cold high-pressure system developing on the east and west sides of Mongolia, generated a dense pressure gradient and strong surface winds exceeding 20 m s<sup>-1</sup> over the GD (Gui et al, 2022). Meanwhile, the early spring snowmelt and low soil moisture in the GD provided favorable conditions for dust to be lifted into the atmosphere. He et al. (2022b) reported a severe dust intrusion event over mainland China, with dust concentrations exceeding 100 μg m<sup>-3</sup> below 2.0 km over Wuhan on 27-30 March 2021. Similar reasons caused the severe dust storm over the GD in the spring of 2023, contributing 77% to the surface dust concentration in southeastern China (Chen et al., 2024, Figure 3 therein). The ratio of DOD to total AOD is presented in Figure 3b. The monthly mean DOD fraction can reach up to 60% during dust-intrusion seasons in spring and winter (e.g., in 2012, 2018, and 2021). In contrast to the pronounced trend in DOD values, the annual mean fraction of DOD fluctuated only between approximately 20% to 30% over the entire period, indicating that the contribution of dust to total AOD remained relatively stable despite year-to-year variations.

## 3.3 Anthropogenic aerosol variation trend

In Hubei and surrounding provinces, wildfires are primarily caused by campfires, smoking, negligent burning by tourists and residents, forestry and agricultural activities, and fireworks (Ying et al., 2018). Natural source fires, such as those induced by lightning are negligible. Therefore, the non-dust component in Wuhan can be primarily attributed to anthropogenic sources. Figure 4a presents the evolution of non-dust AOD over Wuhan from 2010 to 2024, revealing a trend similar to that of total AOD, i.e., a notable decrease trend with a rate of -0.068 yr<sup>-1</sup> (88.3% of the corresponding rate for total AOD) during Stage I, followed by a fluctuating period during Stage II, characterized by a rate of 0.002 yr<sup>-1</sup>. This indicates that the reduction in anthropogenic aerosols has been the dominating factor in the significant improvement of Wuhan's local atmospheric environment. We further estimate the respective contributions of anthropogenic aerosols from the boundary layer (BL) and free troposphere (FT). The BL is defined as the strongly turbulent atmospheric layer directly influenced by dynamic, thermal, and other interactions with the Earth's surface (Peng et al., 2023). Aerosols in the BL predominantly originate from local emissions, while aerosols in the FT are typically from non-local sources via advective transport (Bourgeois et al., 2018). In addition, surface-emitted aerosols can be entrained into the FT through the upward development of convective BL during the daytime. Therefore, partitioning non-dust AOD for the entire atmospheric column into contributions from the BL and FT is conducive to tracing the origins of anthropogenic aerosol from different sources.

Figure 4b shows the evolution of BL and FT non-dust AOD in Wuhan from 2010 to 2024. Over the entire period, the average BL and FT non-dust AOD values are 0.29 and 0.12, respectively, contributing 70.4% and 29.6% to the overall mean non-dust AOD (0.41). In Stage I (2010-2017), the BL and FT non-dust AOD decreased at rates of -0.050 yr<sup>-1</sup> and -0.018 yr<sup>-1</sup>, respectively, suggesting that the BL non-dust AOD was the primary driver for the total AOD reduction (-0.077 yr<sup>-1</sup>), contributing 64.9% to the decline. In Stage II (2018-2024), both the BL and FT non-dust AODs fluctuate with rates of 0.001 yr<sup>-1</sup>.

PM<sub>2.5</sub> has been considered one of the most important air pollutants in BL, with the non-dust components (including organic matter and water-soluble inorganic ions) contributing over 80% in Wuhan (Zhang et al., 2015). To assess air quality in the surface environment of Wuhan, Figure 5a shows the evolution of PM<sub>2.5</sub> concentrations from 2014 to 2024. Similar to the variation of non-dust AODs, the PM<sub>2.5</sub> concentration decreased sharply with a rate of -7.7 μg m<sup>-3</sup> yr<sup>-1</sup> from 2014 to 2017, and then the decline slowed to a rate of -2.6 μg m<sup>-3</sup> yr<sup>-1</sup> from 2018 to 2024. This pattern is consistent with the rates of -4.5 μg m<sup>-3</sup> yr<sup>-1</sup> from 2013 to 2017 and -2.3 μg m<sup>-3</sup> yr<sup>-1</sup> from 2018 to 2020 across China as analyzed by Geng et al. (2024).

Figure 4. (a) Monthly mean non-dust AOD in Wuhan (green dots) from 2010 to 2024. A box plot is presented for each year during 2011-2024. The parameters for box plots are the same as in Figure 2. Gray solid dots represent the annual mean values. (b) Monthly mean BL (dark green dots) and FT (pink dots) non-dust AOD in Wuhan from 2010 to 2024. Linear fitted lines for the periods 2010-2017 and 2018-2024 are represented by solid and dashed lines, respectively.

310

To explain the slower decrease in PM<sub>2.5</sub> concentrations after 2018, Figure 5b presents the concentrations of NO<sub>2</sub> and SO<sub>2</sub>. These two pollutants can chemically transform into sulfate and nitrate, respectively, which together account for 88.3% of the total water-soluble inorganic ions at an urban site in Wuhan (Zhang et al., 2015). Although SO<sub>2</sub> concentrations declined sharply with a rate of -4.5  $\mu$ g m<sup>-3</sup> yr<sup>-1</sup> from 2014 to 2017 due to strengthened industrial emission standards and the upgrading of industrial boilers (Geng et al., 2024), this decline ceased during Stage II (after 2018), with a much slower rate of -0.2  $\mu$ g m<sup>-3</sup> yr<sup>-1</sup>. In contrast, NO<sub>2</sub> concentrations continued to show a slight downward trend with a rate of -1.4  $\mu$ g m<sup>-3</sup> yr<sup>-1</sup>. The NO<sub>2</sub>-to-SO<sub>2</sub> concentration ratio sharply increased from ~ 1.8 in 2014 to ~ 5.3 in 2017, suggesting that NO<sub>2</sub> emissions may have become a more important factor in air pollution during Stage II. Liu et al. (2018) found that the rapid reduction in SO<sub>2</sub> emissions would increase the ammonia concentration, leading to the formation of particulate nitrate, which can offset the effectiveness of emission control measures. In consequence, the imbalance in emission controls for SO<sub>2</sub> and NO<sub>2</sub> may partly explain the slowed decline in PM<sub>2.5</sub> concentrations and the fluctuating trend in non-dust AOD during Stage II. A severe wintertime haze episode linked to nitrate pollution in Wuhan in January 2019 will be analyzed in detail in Section 4.2.

Figure 5. Monthly mean (a) PM<sub>2.5</sub> concentration, and (b) SO<sub>2</sub> and NO<sub>2</sub> concentrations from May 2014 to September 2024. Solid and dashed lines represent the linear fits for the periods 2014-2017 and 2018-2024, respectively. The NO<sub>2</sub>-to-SO<sub>2</sub> concentration ratio is shown in black.

To estimate how meteorological and human emission factors affect the variation of non-dust AOD, the LMG method derived relative contribution and partial correlation analyses of these factors are presented in Figure 6. Apparently, the contribution of meteorological factors has increased from 69.7% during 2014-2017 to 82.4% in Stage II, along with the decrease of effort in PM<sub>2.5</sub> concentration control as mentioned in previous paragraphs. Among meteorological factors, RH contributes the most with 32.9% and 42.4% during the two stages, respectively. Under higher RH conditions, the particle extinction coefficient would greatly increase due to the uptake of water vapor, that is, hygroscopic growth (Miri et al., 2024). The increased contribution of RH indicates a more significant effect of hygroscopic growth in Stage II. The contribution of VV also increased from 1.5% to 8.2%. The positive correlation with AOD indicates that the descending air causes an increase in AOD. Regional pollution can be transported to FT over Wuhan, and the downward wind would bring it into BL and mix with local pollution (Chen et al., 2009). Combined with high RH near the surface, creating an environment that is favour for the formation of secondary organic aerosol (McNeill, 2015). This suggests that regional transboundary pollution might partly offset the effort of local pollution control during Stage II. The contributions of other meteorological factors did not change significantly. It can be noted that the precipitation shows a negative correlation with non-dust AOD due to the removal of aerosols by wet deposition (Figure 6b). However, this negative correlation disappeared in Stage II (Figure 6d), and instead, there is a weak

positive correlation. We suspect that it is because the quality-controlled lidar profile did not strictly correspond to the precipitation time (as mentioned in section 2.6). Frequent precipitation would create a more humid atmospheric environment that is favour for the hygroscopic growth. In addition, the portion of FT non-dust AOD has increased from 27.6% during 2014-2017 to 36.9% in Stage II (as shown in Figure 4b), which can exist above clouds, and may weaken the effect of wet deposition.

**Figure 6.** The LMG method-estimated relative contributions (%) of monthly mean meteorological variations and PM<sub>2.5</sub> concentration on monthly mean non-dust AOD during (a) 2014-2017 and (c) 2018-2024. The partial correlation analyses between non-dust AOD and variations during 2014-2017 and 2018-2024 in Figure 6a (or c) were presented by Figure 6 b and d, respectively. Meteorological parameters are represented by acronyms as: relative humidity (RH), total precipitation (TP), vertical velocity (VV), wind speed (WS), boundary layer height (BLH).

In addition, the annual variation in the LMG method-derived relative contributions of daily mean meteorological and anthropogenic factors are presented in Figure 7. The trends of the dominant factors, RH and PM<sub>2.5</sub>, show increasing and decreasing patterns from Stage I to Stage II, respectively. Overall, the contribution of RH is larger than that of PM<sub>2.5</sub> except in 2015. In Stage II, the notable increase in non-dust AOD in 2018 is associated with the increased contribution of PM<sub>2.5</sub>, which rises to 21% compared with 8% in 2017. In 2019, the contribution of vertical velocity is comparable to that of PM<sub>2.5</sub>, which is inferred to be related to regional pollution transportation. From 2020 to 2021, the contribution of PM<sub>2.5</sub> decreases due to the COVID-19 lockdown. In 2021, wind speed contributes more than PM<sub>2.5</sub>, likely due to enhanced air cleansing by cold air transported from high latitudes under the 'Warm Arctic-Cold Siberia' pattern (Zhang et al., 2021b). After 2022, the contribution of PM<sub>2.5</sub> increases to around 20%, coinciding with the resumption of industrial activities and production. It is worth noting that the contribution of RH is around 60% in Stage II, even exceeds 70% in 2023, which is higher than the value

shown in Figure 6c. This difference arises because the contributions in Figure 7 are based on daily mean values rather than monthly means. Under these circumstances, the daily variability of RH is more pronounced than that of other meteorological variables, and the influence of hygroscopic growth on aerosol backscatter and thus AOD is more direct.

**Figure 7.** The annual variations in the LMG method-estimated relative contributions (%) of daily mean meteorological variations and PM<sub>2.5</sub> concentration on daily mean non-dust AOD.

Figure 8 shows the seasonally averaged profiles of the non-dust extinction coefficient and the monthly variation of BL and FT non-dust AOD from 2010 to 2024. The seasons are defined as spring (March-April-May), summer (June-July-August), autumn (September-October-November), and winter (December-January-February). Approximately 83-88% of anthropogenic aerosols are concentrated below 2 km. The non-dust extinction and mass concentration profiles in summer and autumn are highly consistent, with the largest values observed between 0.7 and 2.5 km with a mean value of 0.12 km<sup>-1</sup> and approximately 83.0 µg m<sup>-3</sup>. Additionally, the highest average non-dust AODs in the BL and FT were observed in June (0.36 and 0.16), followed by September (0.35 and 0.14). One of the major reasons was inferred to be the contribution of ABBS from agricultural areas. In China, agricultural straws are intensively burned during the summer and autumn harvest seasons to enrich cropland nutrients for the next sowing cycle. Based on MODIS fire products, Zha et al. (2013) identified two distinct peaks in agricultural burning: in June (61-86%) and October (5-14%). Several provinces near Wuhan, including Anhui, Henan, Jiangsu, and Shandong, serve as major agricultural heartlands, accounting for over 80% of agricultural fires. As a result, massive ABBS are generated from these agricultural areas within a short period, which can rise into the free troposphere and be transported over long distances (Bourgeois et al., 2018). Using CALIOP aerosol subtype classification data, Lu et al. (2018) also found that central China (109°-116°E and 26°-33°N) experiences the highest frequency of smoke at an altitude of 2 km during summer.

It is worth noting that below 0.7 km, the mean  $\alpha_{nd}$  and  $M_{nd}$  in winter (0.31 km<sup>-1</sup> and 211.8 µg m<sup>-3</sup>) are larger than that in summer (0.27 km<sup>-1</sup> and 185.2 µg m<sup>-3</sup>) and autumn (0.28 km<sup>-1</sup> and 193.1 µg m<sup>-3</sup>). In winter, the high non-dust AODs in the BL were mainly contributed by local anthropogenic aerosols due to the following reasons. First, during winter, low temperatures weaken convective mixing (Miao and Liu, 2019), resulting in a reduced BL height of 0.9 km (yellow dashed line in Figure 8a).

Low temperatures also facilitate the formation of particulate ammonium nitrate from gaseous ammonia and nitric acid, further increasing NH<sub>4</sub>NO<sub>3</sub> concentrations (Han et al., 2008). Second, frequent temperature inversions lead to increased humidity, lower air pressure, and reduced wind speed, which inhibit air circulation and intensify the accumulation of pollutants near the surface (Wu et al., 2014). Third, the concentrations of water-soluble inorganic ions and the relative humidity (71.9%) at the surface are the highest during winter in Wuhan (Zhang et al., 2015), promoting hygroscopic growth, which enhances the particle extinction coefficient and, in turn, AOD.

Figure 8. (a) Profiles of seasonally averaged non-dust extinction coefficient  $\alpha_{nd}$  and mass concentration  $M_{nd}$ . The seasonally averaged boundary layer heights are presented by dashed lines. Box plots of the monthly mean (b) BL and (c) FT non-dust AOD for each month from 2010 to 2024. The parameters for box plots are the same as in Figure 2.

#### 4 Case studies on intense pollution episodes

As discussed in Section 3.3, the seasonal characteristics of anthropogenic aerosols show enhanced extinction coefficients at higher altitudes in summer and near the surface in winter. This reflects two typical aerosol pollution patterns: transboundary aerosol intrusion in summer and local pollution in winter. Massive amounts of ABBS are generated from agricultural straws during the summer harvest season, which can be transported over a long distance to Wuhan, resulting in pronounced extinction features at FT and BL in summer. Here, we present two typical air pollution cases: summertime transboundary ABBS in June

2014 and wintertime local anthropogenic aerosol pollution in January 2019. These cases provide valuable insights into the dominant mechanisms driving the long-term AOD trends.

## 4.1 Summertime biomass burning smoke case in June 2014

A notable agricultural straw burning event occurred in central China and affected the surrounding regions in June 2014 (Wu et al., 2017). Figure 9a shows the hourly surface PM<sub>2.5</sub> and CO concentrations in Wuhan during June 2014, together with the OMPS-measured UVAI between 12 and 14 LT during 9-13 June. On 9 June, the UVAI value was nearly zero, indicating the absence of UV-absorbing aerosols and confirming that the fresh smoke particles had not yet arrived. By 13 June, with the arrival of ABBS, the PM<sub>2.5</sub> and CO concentrations, as well as UVAI values, sharply increased to 479.0 μg m<sup>-3</sup>, 2.1 mg m<sup>-3</sup> and 4.2, respectively, revealing the presence of massive smoke particles. Lidar observations in Wuhan during the period marked by the gray shading in Figure 9a are presented in Figures 9b-k. It should be noted that the lidar system did not operate before 1600 LT on 10 June and after 0400 LT on 12 June. During these three days, the major aerosol layer consistently appeared below altitudes of 3-4 km, with AODs ranging from 1.35 to 2.16. At altitudes above 1.0 km, irregularly shaped smoke particles were observed with the mean  $\delta_p$  of 0.08-0.11, indicating that these particles were transported from the fire area to Wuhan within the free troposphere in a short time and had not yet undergone significant aging. The increase in the mean  $\alpha_p$  from 0.16 to 0.39 km<sup>-1</sup> above 1.0 km between 10 and 12 June further suggests the accumulation of additional smoke particles at higher altitudes, which subsequently mixed downward into the BL. Below 1.0 km,  $\alpha_p$  increased from 1.08 km<sup>-1</sup> on 10 June to 1.31 km<sup>-1</sup> on 12 June, indicating that ABBS in the FT gradually descended and mixed into the BL. From 10 June to the morning of 11 June, the mean  $\delta_p$  of 0.06-0.07 was observed below 1.0 km, indicating that smoke particles mixed with urban/industrial pollutants near the surface and were sufficiently aged. However, in the evening of 11 June, some unaged soot particles descended to near-surface levels, with  $\delta_p$  close to 0.1 at ~0.8 km.

Zhang et al. (2014) reported a severe air pollution event near Wuhan in June 2012, caused by ABBS originating from Anhui Province. This event led to an increased PM<sub>2.5</sub> concentration by nearly an order of magnitude and BC concentration by more than 3 times, compared to normal conditions. The aerosol layers were primarily observed below 1 km, with average particle extinction coefficients as high as 1 km<sup>-1</sup> according to CALIOP observations, which were comparable to those of 0.85-1.31 km<sup>-1</sup> observed in the case presented in this study. On 3 June 2011, another intense ABBS episode occurred in Nanjing (31.5°N, 118.5°E), resulting in an increased AOD of 0.6-3.0 at 500 nm (Wu et al., 2017). Ground-based lidar observations revealed that the major aerosol layers were located below 1.5 km, with significantly enhanced extinction coefficients exceeding 1.0 km<sup>-1</sup>. In general, ABBS from agricultural straw burning during summer were a common phenomenon in the past decades and caused frequent severe air pollution. However, this is no longer the case, according to our long-term lidar monitoring in recent years. This shift can be attributed to the several enforcements of straw burning bans (Huang et al., 2021; Wang et al., 2023b), incentive programs to encourage a comprehensive use of straws, and the development of biomass energy (Sun et al., 2019).

Figure 9. (a) PM<sub>2.5</sub>, CO concentrations at the surface, and UVAI measured by OMPS in Wuhan on 9-13 June 2014. Time-altitude contour plots of (b) range-corrected signal and (c) volume depolarization ratio measured by polarization lidar over Wuhan from 10 to 12 June (local time, UTC+8). The gray shaded area indicates the period of lidar observations. Selected typical profiles of (d-g) extinction coefficient and (h-k) particle depolarization ratio.

To illustrate the actual smoke plumes and prevailing weather conditions, Figures 10a-e present MODIS corrected reflectance and fire/thermal anomalies over the region spanning 29-35°N and 112-118°E during 9-13 June. Most fire hotspots were concentrated in northern Anhui province, generating distinct smoke plumes (gray areas). The two-day backward trajectories starting from Wuhan at an altitude of 1.5 km were simulated each day during 10-12 June using the HYSPLIT model (Figures 10b-d). Due to the lack of lidar observations over Wuhan on 13 June, we instead simulated a backward trajectory initialized at ground level at 0900 LT, based on the peaks of surface-measured PM<sub>2.5</sub> and CO concentrations (see Figure 9a). All trajectories passed through the fire-affected areas, suggesting that the observed aerosol layers over Wuhan were likely associated with ABBS. To intuitively show the spatial extent of the smoke plumes, Figures 10 f-j present the UVAI data. On 10-11 June, ABBS plumes initially generated in Anhui and subsequently dispersed toward the boundary region between Anhui and Henan provinces, with only a small fraction extending to Hubei. After 12 June, most ABBS were transported southwestward toward Wuhan and adjacent regions, resulting in a pronounced UVAI increase near Wuhan (Figure 9a). On 11 June, CALIOP-observed total attenuated backscatter showed that the ABBS plumes were mainly distributed from the surface to an altitude of

5 km. The volume depolarization ratio varied from 0 to 0.2 (blue pixels with occasional yellow ones), indicating a mixture of spherical and non-spherical smoke particles. Relatively high VDR values (>0.15) generally suggest the presence of fresh smoke, which is less oxidized and surface-coated (Haarig et al., 2018). In addition, Figure 10m compares particle extinction observed by CALIOP at 30-31°N with that measured by ground-based polarization lidar located approximately 140 km away. The particle extinction profiles agree well above 2.0 km. However, below 2.0 km, the mean particle extinction from the ground-based polarization lidar (0.71 km<sup>-1</sup>) was significantly higher than that from CALIOP (0.15 km<sup>-1</sup>), reflecting horizontal nonuniformity in aerosol loading, i.e., more severe urban air pollution compared to the relatively clean rural areas (green line in Figures 10c and d).

Figure 10. (a-e) MODIS corrected reflectance and fire/thermal anomalies in central China around 10:30 LT from 9-13 June (MODIS, 2024). Panels (b-d) includes 2-d backward trajectories originating from Wuhan at 1.5 km at 2000 LT on 10-11 June, at 0400 LT on 12 June, and at 30 m at 0900LT on 13 June; (f-j) UVAI measurements from OMPS at 12-14 LT during 9-13 June; CALIOP-observed 532-nm (k) total attenuated backscatter coefficient, (l) volume depolarization ratio, and (m) extinction profile between 30-31°N at 02 LT on 11 June. The extinction profile observed by polarization lidar at 02-04 LT on 11 June was also presented. The CALIOP orbit footprints corresponding to (k) and (l) are shown in (c) and (h), with green lines highlighting the profiles presented in (m).

## 4.2 Wintertime haze case in January 2019





A severe haze episode in January 2019, as shown in Figure 11. The PM<sub>2.5</sub> concentration approximately doubled from the average value of 80 μg m<sup>-3</sup> on 23-24 January to 185 μg m<sup>-3</sup> at night on 25 January. Figures 11b-k show the corresponding lidar

observations during 23-26 January. Aerosols were predominantly concentrated below 1.5 km throughout the four days. Consistent with the increase in PM<sub>2.5</sub> concentrations on 25 January, the AOD increased by a factor of 6.1, from 0.20 to 1.22, over the period from 23-26 January. It is noted that the increase in AOD was more pronounced than that in PM<sub>2.5</sub> levels. Previous studies have shown that an increase in water-soluble inorganic particle fractions, which have been reported as one of the major components in PM<sub>2.5</sub> during haze events in Wuhan (Zhang et al., 2015), facilitates the particle hygroscopic growth (Wang et al., 2020; Hu et al., 2021b), leading to higher  $\alpha_p$  and lower  $\delta_p$  (Zieger et al., 2013; Tan et al., 2019). As seen from Figures 11d-k, on 26 January, a larger mean  $\alpha_p$  of 1.11 km<sup>-1</sup> and a lower mean  $\delta_p$  of 0.05 were derived below 1.0 km, compared to values of 0.17-0.19 km<sup>-1</sup> and 0.11 on 23-24 January, respectively. This suggests a shift in the dominant aerosol type near the surface, from a non-spherical dust and urban aerosol mixture on 23-24 January to hygroscopic and spherical particles by 26 January. For comparison, a similar haze episode in Wuhan in January 2013, reported by Zhang et al. (2021a), exhibited comparable characteristics, including a low  $\delta_p$  of 0.05 (at 532 nm) and a high AOD of 1.32 (at 500 nm).

Figure 11. (a) PM<sub>2.5</sub> concentration in Wuhan on 23-26 January 2019, and time-altitude contour plots of (b) range-corrected signal and (c) volume depolarization ratio measured by polarization lidar over Wuhan during the same period. Selected typical profiles of (d-g) extinction coefficient and (h-k) particle depolarization ratio.

## 5 Summary and conclusions







This study examines the long-term characteristics of tropospheric aerosols over Wuhan from 2010 to 2024, using ground-based polarization lidar observations together with surface pollution monitoring data, ERA5 reanalysis data, radiosonde measurements, and multi-satellite observations. The long-term variation in total AOD can be divided into two phases: a declining trend with a rate of -0.077 yr<sup>-1</sup> during 2010-2017 (Stage I) and a fluctuating period during 2018-2024 (Stage II). Similarly, the surface PM<sub>2.5</sub> concentration decreased rapidly with a rate of -7.7 µg m<sup>-3</sup> yr<sup>-1</sup> during Stage I, followed by a slower reduction of -2.6 µg m<sup>-3</sup> yr<sup>-1</sup> during Stage II. These long-term trends in AOD and PM<sub>2.5</sub> concentrations are consistent with findings from previous studies covering similar periods (de Leeuw et al., 2021, 2022; Geng et al., 2024).

The contribution of AOD was further divided into natural (dust) and anthropogenic (non-dust) aerosol components. For the dust component, the DOD consistently decreased with a rate of -0.011 yr<sup>-1</sup> until August 2020, attributed to reduced surface wind speeds in Asian dust source regions and increased vegetation cover in northwestern China, both of which mitigated the long-range transport of dust aerosols to Wuhan (Jing et al., 2024). However, it is noteworthy that extreme Asian dust storm outbreaks have become more frequent and intense during spring in the past four years (since 2021), due to the exceptionally strong Mongolian cyclone conditions, early spring snowmelt, and low soil moisture in the GD (Gui et al, 2022).

For the anthropogenic aerosol (non-dust) component, its AOD in general exhibited a trend similar to that of total AOD. During Stage I, the non-dust AOD contributed 88.3% to the reduction in total AOD (-0.068 yr<sup>-1</sup> versus -0.077 yr<sup>-1</sup>). This reduction was primarily contributed by the decrease in BL AOD (contributing 64.9% to total AOD), highlighting the significant impact of reduced emissions of local anthropogenic aerosols due to effective government policies. During Stage II, the non-dust AOD showed a fluctuating trend. The imbalanced control of  $SO_2$  and  $NO_2$  emissions led to a sharp rise in the  $NO_2$ -to- $SO_2$  concentration ratio, from approximately 1.8 in 2014 to 5.3 in 2017, which may have partially offset the pollution control efforts during Stage II. Through the LMG method, the relative contribution of meteorological factors to non-dust AOD variation has increased from 69.7% during 2014-2017 to 82.4% in Stage II. Among them, RH and VV play a more important role in Stage II, indicating a more significant effect of hygroscopic growth and regional transboundary pollution. The seasonal characteristics of anthropogenic aerosols were also analyzed. In summer and autumn, anthropogenic aerosols showed enhanced extinction coefficient (0.12 km<sup>-1</sup>) and mass concentration (83.0 µg m<sup>-3</sup>) at altitudes of 0.7-2.5 km, mainly due to smoke produced by agricultural-straw burning. In contrast, anthropogenic aerosols in winter were generally concentrated below 0.7 km, with much higher  $\alpha_{nd}$  and  $M_{nd}$  of 0.31 km<sup>-1</sup> and 211.8 µg m<sup>-3</sup>, respectively.

In addition, two typical air pollution case studies were presented: summertime transboundary ABBS in June 2014 and wintertime local anthropogenic aerosol pollution in January 2019. From 9-13 June 2014, ABBS plumes originating from agricultural straw burning in Anhui Province were transported to Wuhan. The smoke layer was generally located below 4 km, with AOD values of 1.35-2.16. Within the BL, high aerosol concentrations were observed below 1 km, with  $\alpha_p$  values of 0.85-1.31 km<sup>-1</sup> and  $\delta_p$  values of 0.06-0.07, suggesting the predominance of aged smoke particles. In January 2019, a severe haze event occurred in Wuhan, with PM<sub>2.5</sub> concentrations reaching up to 185 µg m<sup>-3</sup> on 25 January. Meanwhile, the AOD

increased by a factor of 6.1, from 0.20 on 23 January to 1.22 on 26 January. On 26 January, a significant large  $\alpha_p$  of 1.11 km<sup>-1</sup> <sup>1</sup> and relatively low  $\delta_p$  of 0.05 were recorded below 1 km, suggesting the accumulation of spherical anthropogenic aerosols. Our previous work (Yin et al., 2021b) took advantage of polarization lidar observations to examine the long-term variations in the optical properties of tropospheric aerosols over Wuhan from 2010 to 2020, revealing a downward trend in tropospheric AOD. The present study, as an extension of Yin et al. (2021b), updates the observational results to include the most recent four years and reports a halt in the reduction of AOD since 2018. This updated analysis promotes our understanding of how Wuhan's atmospheric environment responds to the government's anthropogenic emission control policies and natural mineral 515 dust activities in East Asian desert regions. Due to the limitations of single-wavelength polarization lidar, distinguishing smoke aerosols from locally emitted urban aerosols is only feasible by incorporating large-scale spaceborne observations and simulations of air mass backward trajectories. In the future, we plan to involve observations from pure rotational Raman lidars at both 355 nm and 532 nm (Pan et al., 2020; Liu et al., 2019; Yi et al., 2024). This will enable the exploration of the relationship 520 between lidar ratio and particle depolarization ratio (Peng et al., 2021; Floutsi et al., 2023), facilitating the classification of aerosol types. In addition, the particle extinction coefficient is known to increase significantly due to the hygroscopic growth of aerosols under high humidity conditions (Zieger et al., 2013). This phenomenon will be analyzed in future studies to further evaluate the long-term variation in the dry optical properties of tropospheric aerosols over Wuhan.

## Data availability

CALIOP data can be obtained from https://subset.larc.nasa.gov/ (CALIPSO, 2025). OMPS data can be obtained from https://www.earthdata.nasa.gov/sensors/omps (OMPS, 2025). MODIS daily corrected reflectance imageries can be obtained from https://worldview.earthdata.nasa.gov/ (MODIS, 2025). ERA5 reanalysis data can be obtained from https://cds.climate.copernicus.eu/datasets (ERA5, 2025). The air pollution monitoring data can be obtained from http://www.cnemc.cn. The radiosonde data can be obtained from http://weather.uwyo.edu/upperair/sounding.html. The HYSPLIT model is available at https://www.arl.noaa.gov (HYSPLIT, 2025). Lidar data used to generate the results of this paper are available from the authors upon request (e-mail: yf@whu.edu.cn).

## **Author contributions**


YH, DJ, and ZY analyzed the data and wrote the manuscript. ZY, KH, and FL participated in scientific discussions and reviewed and proofread the manuscript. YH and FY conceived the research and acquired the research funding. FY led the study.

## **Competing interests**

The contact author has declared that none of the authors has any competing interests.

## Financial support

This work was supported by the National Natural Science Foundation of China (grant nos. 42575138, 42005101, 41927804, and 42205130), the Hubei Provincial Natural Science Foundation of China (2023AFB617), Hubei Provincial Special Project for Central Government Guidance on Local Science and Technology Development (2025CFC003), and the Meridian Space Weather Monitoring Project (China).

#### Acknowledgements

The authors thank the colleagues who participated in the operation of the lidar system at our site. We also acknowledge the
Atmospheric Science Data Central (ASDC) at the National Aeronautics and Space Administration (NASA) Langley Research
Center for providing the CALIPSO data, NASA/National Oceanic and Atmospheric Administration (NOAA) for the OMPS
data, NASA Earth Science Data and Information System (ESDIS) project for MODIS daily corrected reflectance imageries,
the European Centre for Medium-Range Weather Forecasts (ECMWF) for ERA5 reanalysis data, the China National
Environmental Monitoring Center (CNEMC) for Air pollution monitoring data, the University of Wyoming for radiosonde
data, and the NOAA Air Resources Laboratory (ARL) for the HYSPLIT model.

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
