# Peer review of "Evolution of tropospheric aerosols over central China during 2010-2024 as observed by lidar"

_EGUsphere, 2025_

## Author Comment (AC1)

**Responses to RC1**

**General Remarks**

My ratings are based on a balance between the main part of the MS (very good, minor revision) and the descriptions of the case studies (poor, major revision).

The authors present a study on the evolution of aerosols over Wuhan during 2010-2024. The study is based on the use of a polarization lidar providing information on the vertical structure of aerosol particles and their polarization state as a source of information on aerosol sphericity to discriminate between dust and non-dust aerosol, where dust is considered natural and non-dust is considered anthropogenic. In support, they use a comprehensive set of data including satellite observations, ground-based monitoring data, air mass back trajectories, radiosonde data and meteorological information. Considering the variation of the lidar-derived AOD, the authors split the time series into two stages. During stage I (2010-2017) the AOD decreases, during stage II (2018-2024) the AOD is on average constant. The variations of the AOD are discussed, reasons for the decrease in Stage I are presented and fluctuations in Stage 2 are discussed. A similar analysis is made for dust and non-dust aerosol and for the latter also at two levels: in free troposphere and in the total atmospheric boundary layer. Time series of aerosol mass concentrations PM2.5 and aerosol precursor gases NO2 and SO2 are presented in support of suggestions of changes in aerosol chemical composition. In addition, two case studies are presented.

The manuscript is overall well-written and the analysis is interesting. However, I have some comments, questions and suggestions for further improvement which need to be addressed before it can be accepted for publication in ACP. In particular, while the main part of the MS is quite clearly presented (minor revision), two case studies have been added which are much less clear and need much clarification (major revision).

**Response:** We sincerely appreciate your thorough review and valuable comments in this round, which have greatly contributed to improving our work. We have clarified the definition of the two periods with distinct variation patterns in AOD and dust optical depth (DOD), and we have added further analyses and discussions on the causes of the observed 'two-stage' variation patterns over the past 15 years. In particular, the statements in the two case studies have been substantially modified to make them much clearer and more easily understood by readers. Point-by-point responses are provided, and the manuscript has been revised accordingly.

**Specific Comments:**

**Comment:** The main subject of this MS is the decrease of the AOD in 2 different periods. However, there is no discussion on how these different periods were determined.

**Response:** Thank you very much for pointing this out. Based on the long-term variation of monthly AOD, two stages can be preliminarily identified: an evident declining phase (Stage I) and a fluctuating phase (Stage II), with the transition occurring around 2017-2018. To determine the transition point between these two stages, we tested different scenarios by shifting the transitional month (i.e., the end of Stage I) from January 2017 to December 2018. The corresponding slopes of AOD variations for Stage I and II are presented in Figure 1R. Ideally, the transitional month would yield both the steepest decline in Stage I and a slope that is closest to zero in Stage II; however, identifying the optimal choice requires balancing between these two criteria. As illustrated in Figure 1R, the transition most likely appeared within the first 2-3 months of 2018. Therefore, for clarity and consistency, we adopt December 2017 as the end of Stage I and January 2018 as the beginning of Stage II. Relevant descriptions have been added in Section 3.1. (please see lines 187-193)

[Figure]

Figure 1R. Slopes of AOD variations for stages I and II by using each month from Jan. 2017 to Dec. 2018 as the transitional month.

**Comment:** Stage I is from 2010 to 2017, but for DOD it ends 2020. However, the DOD variation in stage II 2017-2024 is very small (smaller than for total AOD in stage II), so what is the motivation to extend the DOD fit to 2020?

**Response:** We do not subjectively determine the transitional point between the two stages of DOD variation. Slopes calculated by shifting the end month of DOD fit from January 2017 to September 2024 are presented in Figure 2R. When the DOD fit is terminated between 2017 and 2020, the resulting slopes remain around -0.010 $yr^{-1}$, indicating that the downward trend persisted during that period. It should be mentioned that in Figure 2R, the large fluctuation in DOD slope is primarily driven by pronounced seasonal patterns over Wuhan; nevertheless, a clear divergence between the two periods, with the end of 2020 as the transition point, is evidently seen. However, as the end of the DOD fit extends to months after 2021, the slope rapidly approaches zero. Therefore, we choose to extend the DOD fit to the end of 2020. The different transitional time for total AOD and DOD, derived from long-term lidar observations, is also a clear indication that the main characteristics of AOD over Wuhan are not controlled by natural sources.

We have added some sentences regarding the determination of the end of the downtrend for long-term DOD variation in the revised manuscript. (please see lines 251-253)

[Figure]

Figure 2R. Slopes of DOD variations for downtrend using each month from 2017 to 2024 as the end of the stage.

**Comment:** Another question is that for both total AOD and DOD a maximum is observed in 2014 and AOD was high and variable before the decline in 2014. The Clean Air Action plan started in 2013. Would it be more logical to start the fit in 2013? Does it make a difference?

**Response:** For AOD, the annual mean values in 2011 (0.89) and 2012 (0.83) were higher than those in 2014 (0.72). The exceptionally high monthly mean AOD of 1.18 in 2014 was caused by transboundary biomass burning smoke intrusion, as discussed in the case study in Section 4.1. If excluding this extreme case, the annual mean AOD in 2014 will decrease to 0.65, which is comparable to the 2013 value of 0.60. Therefore, the 2014 AOD still aligns with the long-term downward trend. Additionally, several air pollution control policies were implemented over the past 15 years to promote sustainable development. For instance, the "*Technical guidelines for air pollution control projects*" (Ministry of Ecology and Environment of the People's Republic of China, 2010) was issued at the end of 2010 and came into effect in 2011. The "*Air Pollution Prevention and Control Action Plan*" in 2013 was one of the most impactful policies. If adopting 2013 as the starting year of the fit, the slope of AOD in Stage I was -0.072 yr$^{-1}$, which is slightly less than -0.077 yr$^{-1}$ obtained when starting from 2010. As a result, it is more reasonable to regard the long-term declining trend as beginning with the first year of our observations, i.e., 2010.

For DOD, dust emissions are mainly influenced by dust activities and meteorological factors in desert regions, rather than local urban pollution. As noted in section 3.2, recent extreme springtime dust events have been linked to usually strong Mongolian cyclone conditions, early spring snowmelt, and low soil moisture in the Gobi Desert (Gui et al, 2022; Chen et al., 2024).

**Comments:** The case studies need to be better explained.
1.  In particular, the authors start with Fig. 7 with only information on the major event, on 13 June, in Fig. 7j. It would be more logical when Fig 8a would be shown as introduction to the event, followed by clear explanations of all other Figures presented.
2.  In addition, the one air mass trajectory shown does not arrive on at the time of the peak PM, why are not more trajectories shown?
3.  That one trajectory indicates an anticyclonic circulation, does that explain the day-to-day UVAI pattern? And the highest UVAI NW off the burnt area on 6.11 and south on 6.12 and the intensification and elongated pattern in 6.13?

**Response:**
1.  Thank you for the reviewer's suggestion. We have reversed the order of the original Figures 7 and 8, re-numbering them as Figures 8 (lidar observation) and Figure 9 (spaceborne measurements). The corresponding descriptions in Section 4.1 have been modified accordingly.
2.  According to the reviewer's suggestion, HYSPLIT backward trajectories originating from Wuhan for each day during 10-13 June have been added to Figures 9b-e. The statements have been revised to "**The two-day backward trajectories starting from Wuhan at an altitude of 1.5 km were simulated each day during 10-12 June using the HYSPLIT model (Figures 9b-d). Due to the lack of lidar observations over Wuhan on 13 June, we instead simulated a backward trajectory initialized at ground level at 0900 LT, based on the peaks of surface-measured PM$_{2.5}$ and CO concentrations (see Figure 8a). All trajectories passed through the fire-affected areas, suggesting that the observed aerosol layers over Wuhan were likely associated with ABBS.**" (please see lines 412-417)
3.  We have updated the backward trajectories and the associated analysis, and anticyclonic circulation is no longer involved. The backward trajectories of air masses originating from Wuhan passed through the fire-affected regions, indicating that the lidar-observed elevated aerosol plume over Wuhan likely originated from transboundary smoke, as supported by the height and time information of the lidar observations. While day-to-day UVAI variations reflected the dispersion of smoke plumes from the fire region, leading to enhanced UVAI values in Wuhan and the surrounding areas. Therefore, the combination of simulated backward trajectories and satellite-measured UVAI provides valuable evidence for inferring the origins and

compositions of the aloft aerosol layers observed in the free troposphere over Wuhan. For instance, the trajectories starting on 11 and 12 June (Figure 9c and d) traced back to areas of high UVAI on 10 and 11 June, respectively (Figure 9g and h). Similarly, the trajectory starting on 13 June approximately coincided with the intensification and elongation of the UVAI pattern on that day. The UVAI evolution has been described in detail as follows. "**On 10-11 June, ABBS plumes initially generated in Anhui and subsequently dispersed toward the boundary region between Anhui and Henan provinces, with only a small fraction extending to Hubei. After 12 June, most ABBS were transported southwestward toward Wuhan and adjacent regions, resulting in a pronounced UVAI increase near Wuhan (Figure 8a).**" (please see lines 417-420)

[Figure]

Figure 3R. (Figure 9 in revised manuscript) (a-e) MODIS corrected reflectance and fire/thermal anomalies in central China around 10:30 LT from 9-13 June (MODIS, 2024). Panels (b-d) includes 2-d backward trajectories originating from Wuhan at 1.5 km at 2000 LT on 10-11 June, at 0400 LT on 12 June, and at 30 m at 0900LT on 13 June; (f-j) UVAI measurements from OMPS at 12-14 LT during 9-13 June; CALIOP-observed 532-nm (k) total attenuated backscatter coefficient, (l) volume depolarization ratio, and (m) extinction profile between 30-31°N at 02 LT on 11 June. The extinction profile observed by polarization lidar at 02-04 LT on 11 June was also presented. The CALIOP orbit footprints corresponding to (k) and (l) are shown in (c) and (h), with green lines highlighting the profiles presented in (m).

**Comments:** In more detail:

1. Fig. 7 a-e: Why is the MODIS corrected reflectance shown? Or is the burnt area (red spots) the most important? Figs a-e show the haze but also the clouds in 3 out of 5 scenes (a, d, e) which, in e obscure all relevant info. What does that mean for the UVAI? Is it measured only above the clouds? Fig e does not show the burnt area because of the cloud cover.

2. Figs f-j indeed show the increasing UVAI, but, in view of the clouds, what is the relation with the surface concentrations in Fig 8a on 6.13? The variation of UVAI in Fig 8a shows remarkable good agreement with PM2.5, likely because it was scaled this way, but in view of the clouds this does not seem credible.

3. Why is Fig 7I shown? What does it tell us? Depol is very small.

**Response:**

1. MODIS corrected reflectance imagery was used to display the actual smoke plumes (in gray) together with prevailing weather conditions. Fire and thermal anomalies were used to indicate the specific fire locations.

UVAI was applied to show the absorbing aerosols that intermingled with or above clouds (Torres et al., 2007). The combination of these three datasets enables the identification of fire sources, as well as the diffusion and transport of fire-emitted smoke plumes. The revised statements are as follows. "**To illustrate the actual smoke plumes and prevailing weather conditions, Figures 9a-e present MODIS corrected reflectance and fire/thermal anomalies over the region spanning 29-35°N and 112-118°E during 9-13 June**" (please see lines 413-414) and "**To intuitively show the spatial extent of the smoke plumes, Figures 9f-j present the UVAI data.**" (please see line 420) Additional descriptions of MODIS and UVAI have been included in Section 2.2. (please see lines 118-138)

2. As mentioned in the previous response, UVAI represents absorbing aerosols that are mixed with or located above clouds; thus, it has no direct relationship with $PM_{2.5}$. The observed increases in both were driven by smoke transported to Wuhan and descended to the surface, as shown by ground-based lidar, multiple satellite observations, and trajectory simulations.

3. Figure 4R shows a local pollution event near Wuhan observed by CALIOP, characterized by enhanced backscatter coefficient concentrated below 3 km and low VDR values of 0-0.1 (in blue), indicating the presence of spherical particles. In comparison, the VDR in our case study ranged from 0-0.2, represented by alternating yellow (0.1-0.2) and blue (0-0.1), suggesting a mixture of spherical and non-spherical particles. Relatively high VDR values (>0.15) are indicative of fresh smoke that is less oxidized and surface-coated (Haarig et al., 2018). This result supports the interpretation that smoke plumes were transported over a short distance from adjacent provinces to Wuhan. Nevertheless, VDR alone is not a reliable indicator of particle aging, as it also includes contributions from molecular depolarization. The particle depolarization ratio (PDR) provides a more robust measure and was further adopted (using ground-based lidar observations as shown in Figure 8) to analyze the aging level of smoke particles. For this reason, we prefer not to overinterpret the CALIOP-derived VDR here. The revised description is as follows. "**On 11 June, CALIOP-observed total attenuated backscatter showed that the ABBS plumes were mainly distributed from the surface to an altitude of 5 km. The volume depolarization ratio varied from 0 to 0.2 (blue pixels with occasional yellow ones), indicating a mixture of spherical and non-spherical smoke particles. Relatively high VDR values (>0.15) generally suggest the presence of fresh smoke, which is less oxidized and surface-coated (Haarig et al., 2018).**" (please see lines 420-424)

[Figure]

Figure 4R. CALIOP observed (a) total attenuated backscatter and (b) volume depolarization ratio of polluted continental aerosols near Wuhan at 1810-1823 UTC on 21 August 2017.

**Comments:**
1. Figure 7 shows an air mass trajectory but why for only 1 day? Why not for every day? Or several times on a day? And what does it show? It went back from Wuhan, at 2000LT, and as indicated by dates (m.d), it arrived on the 11th? It probably started low, on the 9th, suddenly was lifted to 3 km and stayed there all day on the 10th and dropped again very fast back to 1 km on the 11th. Is that how we should read it? And then we should look at the other maps when the trajectory overpasses a burnt area and may have picked up smoke.

2. The maps indicate it was on the 10th, between about 33 and 34 N, and the CALIOP track (what was the overpass time? Ascending or descending? What are the red lines at the bottom of the CALIOP Figures? Please mention) shows high AOD between 32 and 34 N but at the east of the air mass trajectory, and UVAI

(when was OMPS overpass LT?) was not very high under the CALIOP overpass (Fig. g). CALIOP has crossed the air mass trajectory but a little later. The air mass passed over the burnt area on 6.10 and may have picked up some smoke, but UVAI shows a relatively small signal. The green lines in the CALIOP trajectory indicate smoke presence but UVAI is not enhanced.

3. Fig. g to f suggests that smoke had been transported south and arrived over Wuhan. Why are Figures shown for 12 and 13 June?

4. In the text: L319-320 burnt area over northern Anhui, so why is smoke formed in Henan? How does that relate to the air mass trajectory and the CALIOP and OMPS observations?

**Response:**

1. In light of the reviewer's comments, more HYSPLIT-simulated backward trajectories (one trajectory simulated for each day) have been added in the updated Figure 9. Please recheck them.

2. For clarity, we have added the specific time (convert to local time, i.e., UTC+8) of the day in the title of subfigures for MODIS, OMPS, and CALIOP in the updated Figure 9. The overpass time of the CALIOP track is at ~02LT on 11 June (~18 UTC on 10 June), corresponding to the descending phase at nighttime. The red lines at the bottom of the CALIOP Figures indicate the ground levels. It can be noted that the overpass time of CALIOP is between the UVAI measurements on 10 and 11 June (Figures 9 g and h). The CALIOP orbit footprint overlaps with the strongly enhanced UVAI in northern Anhui and slightly enhanced UVAI in eastern Hubei (Figure 9h). The updated green line on the CALIOP orbit footprint represents the CALIOP measured extinction profile presented in Figure 9m. We have also added a detailed description of satellite observation in the revised manuscript. (please see lines 410-429)

3. Lidar observed the aloft aerosol layer from ground to ~3.5 km over Wuhan on 12 June, and surface $PM_{2.5}$ measurement suggested the existence of high aerosol load near the surface of Wuhan. These observations indicate that smoke aerosol continuously impacted Wuhan on 12-13 June. This is why we also provided the Figures for these two days.

4. Smoke plumes were initially generated in Anhui and subsequently dispersed toward the boundary region between Anhui and Henan. Thank you for pointing out these contradictory statements. Relevant statements have been revised as follows. **"ABBS plumes initially generated in Anhui and subsequently dispersed toward the boundary region between Anhui and Henan provinces, with only a small fraction extending to Hubei."** (please see lines 417-419) We have clarified the combined observation of MODIS OMPS and CALIOP in previous responses, and the relevant revisions have been presented. (please see lines 410-429)

**Comment:** In other words: guide the reader and explain what is important!

**Response:** Thank you very much for the reviewer's valuable comments, in particular regarding the case studies. For clarity, we have added more detailed explanations in the revised manuscript (please see lines 410-429), as can also be seen from the responses to the comments above.

**Comment:** Likewise, guide us through Fig. 8. The first thing I wonder about is that PM2.5 and CO peak in the evening of 12 June. So why was the air mass trajectory not calculated for the peak PM2.5 and CO time? That would show the sources, right?

**Response:** Thank you very much for the valuable suggestion. More simulated backward trajectories during 10-13 June have been added. The backward trajectory passed through the fire-affected areas, suggesting that the observed aerosol layers over Wuhan were likely associated with the fire-emitted smoke particles. Relevant statements have also been revised in the manuscript accordingly. (please see lines 412-417)

**Comments:**

1. Another is that CALIOP overpass is early in the afternoon and your lidar observations started at 16:00 LT,

both on the 10th. Why is there no comparison?

2. There is a description between L324 and L339, but I miss guidance. For instance, L333-334; "severe air pollution was observed": when (date, time, height), etc. Fig b shows me that BLH was about 1.5 km at 16:00 on 11.6, but the temperature profiles was at 20:00, when BLH had dropped to about zero. And depol at that time was close to 0.1 (red) in the BL. (in contrast to L336) and how do these contrasting statements and observation reflect the next sentence? (L336-339).

3. Further questions are how the extinction, and RH profiles are used? These profiles show clear gradients and transitions which may indicate different layers and may be connected with backscatter profiles in Fig 8b. The authors mention high RH. Which however is only shown in Fig 8e and 8g in a thin layer just above the surface. Further above the RH is so low that little or no hygroscopic growth can be expected. And also PDR does not show correlation with the RH variation.

**Response:**

1. On 11 June, extinction coefficient profiles from CALIOP near Wuhan at 02 LT and polarization lidar measurements during 02-04 LT have been included in updated Figure 9m of the revised manuscript. We have also added the following text regards the comparison "**In addition, Figure 9m compares particle extinction observed by CALIOP at 30-31°N with that measured by ground-based polarization lidar located approximately 140 km away. The particle extinction profiles agree well above 2.0 km. However, below 2.0 km, the mean particle extinction from the ground-based polarization lidar (0.71 km$^{-1}$) was significantly higher than that from CALIOP (0.15 km$^{-1}$), reflecting horizontal nonuniformity in aerosol loading, i.e., more severe urban air pollution compared to the relatively clean rural areas (green line in Figures 9c and d).**" (please see lines 424-429)

2. Thank you for pointing this out. The statement has been revised to "**Below 1.0 km, $\alpha_p$ increased from 1.08 km$^{-1}$ on 10 June to 1.31 km$^{-1}$ on 12 June, indicating that ABBS in the FT gradually descended and mixed into the BL.**" (please see lines 389-390) We previously overlooked the enhanced volume depolarization ratio in the afternoon of 11 June, which led to a PDR close to 0.1 at 0.8 km during the night of 11 June (purple line in Figure 8j). Accordingly, we have revised the previous conclusion that smoke particles near the surface were fully aged. The related statements have been revised to "…**From 10 June to the morning of 11 June, the mean $\delta_p$ of 0.06-0.07 was observed below 1.0 km, indicating that smoke particles mixed with urban/industrial pollutants near the surface and were sufficiently aged. However, in the evening of 11 June, some unaged soot particles descended to near-surface levels, with $\delta_p$ close to 0.1 at ~0.8 km.**" (please see lines 390-393) In addition, as BLH and temperature were not very relevant, we have removed them from Figure 8 and the corresponding discussions.

3. The explanation regarding RH is overly complicated. Since it does not affect the main conclusion of this work, we have decided to remove the related statements. Descriptions of extinction and PDR have been revised accordingly in the manuscript. (please see lines 385-393)

**Comment:** In summary: there is a lot to be said about Figs 7 and 8, please explain.

**Response:** More detailed explanations have been added in responses to the comments above.

**Comments:**

1. And this also applies to the haze case in Sect. 4.2. Figs 9d-k are not explained or used, so why are they presented.

2. Figs a-c are used but attributing the increase in PM only to NO3 formation seems a big step. Why did that not happen on the previous days? Rather, the decrease in NO2 in the evening of the 25th, as opposed to increase of nocturnal NO2 due to chemical reactions, could also be due to a change in transport from areas with smaller NO2 emissions.

**Response:**

1. The descriptions related to Figures 10d-k in the manuscript were as follows *"on 26 January, a larger mean $\alpha_p$ of 1.11 km$^{-1}$ and a lower mean $\delta_p$ of 0.05 were derived below 1.0 km, compared to values of 0.17-0.19 km$^{-1}$ and 0.11 on 23-24 January, respectively."* For clarity, "**As seen from Figures 10d-k**" has been added in the revised manuscript. (please see lines 446-447) To avoid complicated explanations and misleading, RH profiles and BLH have been removed in the revised manuscript.

2. The purpose of presenting the haze event is mainly to show the typical vertical distribution and optical properties (e.g., extinction and depolarization ratio) of winter urban pollution over Wuhan, based on our remote sensing measurements. Since we did not have comprehensive atmospheric chemical measurements, providing additional explanations would involve a risk of overinterpretation. To ensure rigor, we have removed analyses of $NO_2$ and $O_3$ from Figure 10a and the corresponding text in the revised manuscript. Only explanations and analyses of aerosol optical properties are retained, and the related statements have been revised accordingly. (please see lines 439-451)

**Detailed suggestions**

L42: In most of the para the authors discuss AOD, and the slow-down in the decrease of AOD, which is also discussed further below (L165). Therefore, it seems a bit strange that on L42 the slowdown in PM2.5 is mentioned, which has been reported to respond differently than AOD.

**Response:** AOD and PM$_{2.5}$ concentrations are both important indicators of environmental quality. The PM$_{2.5}$ concentrations are closely linked to human activities near the surface, whereas AOD reflects the aerosol loading throughout the entire atmospheric column. The observed slowdown in PM$_{2.5}$ concentration reduction in China illustrates the weakening of emission control measures, which leads to the central question of this study: Do weakened emission control measures affect AOD over Wuhan? For clarity, we have highlighted our motivation as follows "**In our previous study, we reported a consistent downward trend in AOD during 2010–2020 (Yin et al., 2021b). However, it appears that this decline in AOD ceased and even slightly reversed after 2018 (Yin et al., 2021). Given that the reduction in surface PM$_{2.5}$ concentrations also slowed after 2018 (Geng et al., 2024), it remains unclear whether the consistent downward trend in AOD merely decelerated or completely halted at some point after 2018. In addition, in the additional four years following 2020, several factors including industrial shutdowns during the COVID-19 pandemic, abnormal Asian dust events (Gui et al., 2022; He et al., 2022b), and extreme precipitation (Wang et al., 2023a; Li et al., 2024), may have further influenced AOD levels in Wuhan. Therefore, it is of great interest to extend the analysis by incorporating more recent datasets from our polarization lidar observations to assess the impact of the weakened emission control measures on AOD over Wuhan.**" (please see lines 46-53)

L90 "Note that $\alpha_p$ values below 0.3 km were assumed equal to that at 0.3 km, possibly causing an uncertainty of <0.05 in AOD". Fig. 6 shows seasonally averaged profiles with strong gradients above 300m. Would it be more logical to linearly interpolate to the surface?

**Response:** We appreciate your thoughtful review. Since this work builds on our previous studies (Yin et al., 2021; Jing et al., 2024), we consider it more appropriate to maintain the same data processing approach for consistency. A similar method was also adopted by Baars et al. (2017), who assumed height-independent extinction below the altitude with a complete field-of-view.

Linearly interpolating the extinction profile to the surface would introduce outliers with unrealistically large or even negative extinction coefficients. Thus, both approaches inevitably induce uncertainties in AOD. Besides, as there are only 10 vertical bins within the lowest 0.35 km, their influence on the column-integrated AOD (0-7 km) is limited, with an associated uncertainty of <0.05. Additionally, this study focuses on the long-term evolution of AOD based on large datasets, and the extrapolation method applied to the low-altitude with incomplete FOV does not affect the main conclusions of this work.

L100    suggest to change policy to protocol

**Response:** "policies" has been modified to "protocols".

L174    attributed

**Response:** "attributing" has been modified to "attributed".

L190    Figure 2a caption "fewer than 15 cloud-free profiles being recorded in a given month" why this restriction to 15 profiles? It is not easy to see in the Figure whether the extinction profiles are monthly averaged. Or is that because you use monthly mean data in Fig b?

**Response:** "…**the particle extinction coefficient…**" in Figure 2 caption has been revised to "…**the monthly mean particle extinction coefficient…**".

Here, the pie plot of the number of profiles in each month during 2010-2024 is shown in below. Months with <15 profiles account for only 3% of the total 155 months with valid data. Therefore, if the number of profiles in a certain month is less than 15, it indicates that the statistical results for that month are not representative.

[Figure]

Figure 5R. Distribution of valid aerosol profile numbers in each month.

L 222    Fig 3 caption: change to "DOD for monthly mean" and "The dark orange line".

**Response:** "The long-term evolution of" has been deleted, and "Dark orange line" has been modified to "The dark orange line". The same content in Figures 3 and 4 has also been modified.

L235 "In Wuhan, the non-dust component is primarily attributed to anthropogenic aerosols". In the case studies you show the effect of biomass burning aerosol (anthropogenic). Attributed to straw burning. To your knowledge, are there wildfires in the area contributing to the aerosol content, i.e. which are not anthropogenic?

**Response:** There are three major types of biomass burning: forest fire, agricultural straw open burning, and fuel combustion (Chen et al., 2017). Among these, forest fires can generate natural wildfire smoke. According to the study of forest fire characteristics in China by Ying et al. (2018), forest fires in Hubei and surrounding provinces (roughly marked by a blue circle) are mainly caused by recreational activities, such as campfires, smoking, or negligent burning by tourists and residents. Forest fires from industrial sources (e.g., forestry or agricultural activities) and cultural practices (e.g., fireworks and firecrackers) play a secondary role, while lightning-induced fires are essentially absent. Therefore, the non-dust component in Wuhan can be primarily attributed to anthropogenic aerosols. The corresponding statements in the manuscript have been revised as follows. "**In Hubei and surrounding provinces, wildfires are primarily caused by campfires, smoking, negligent burning by tourists and residents, forestry and agricultural activities, and fireworks (Ying et al., 2018). Natural source fires, such as those induced by lightning are negligible. Therefore, the non-dust component in Wuhan can be primarily attributed to anthropogenic sources.**" (please see lines 270-272)

[Figure]

Figure 6R. Patterns of fire cause at the county level in China (Ying et al., 2018).

L237    notably attributed
**Response:** "a notable" has been modified to "notably attributed".

L240-245    The ABL develops throughout the day and aerosols are usually mixed throughout the ABL, with a gradient across the inversion at the top of the BL. Were the BLH be determine from the individual lidar profiles? If not, how was BLH determined?
**Response:** The BLH data were obtained from ERA5 reanalysis, derived from the ECMWF Integrated Forecasting System's turbulent diffusion and turbulent orographic form drag schemes. Specifically, the bulk Richardson number ($R_i$) method was used to calculate BLH, where $R_i$ represents the ratio of turbulence generated by buoyancy to that generated by mechanical shear (Seidel et al., 2012). The boundary layer height is defined as the lowest level at which the bulk Richardson number reaches the critical threshold of 0.25 (ECMWF, 2017).

L261    Fig 4 caption: "as in Figure 2"
**Response:** "to" has been modified to "as in". The same content in Figures 3 and 6 has also been modified.

L 267 & 270    concentrations
**Response:** "concentration" has been modified to "concentrations".

L 273    effectiveness of emission control measures?
**Response:** "the effectiveness of control" has been modified to "the effectiveness of emission control measures".

L281    Profiles are shown in Fig. 6a
**Response:** The original sentence is a general summary of Figure 6, not only profiles, but also the monthly variation of BL and FT non-dust AOD.

L288    combusted or combustion, change to burned or burning (here and all other occurrences)
**Response:** "combusted" and "combustion" have been modified to "burned" and "burning".

L307    Fig. 6 caption: seasonally averaged; the same as in Figure;

**Response:** Three places of "seasonal average" have been modified to "seasonally averaged".

L325   UVAI is zero and indicates the presence of UV-absorbing aerosols: is that a typo?
**Response:** "presence" has been modified to "absence".

L406   fluctuated with a rate of 0.002 /yr. That would mean that AOD increased with that rate and varied around that line by +/- 0.2
**Response:** "the non-dust AOD fluctuated with a rate of 0.002 yr$^{-1}$" has been modified to "the non-dust AOD showed a fluctuating trend".

L416   "plenty of aerosols", do you mean "much aerosol" or "high aerosol concentrations?"
**Response:** "plenty of aerosols" has been modified to "high aerosol concentrations".

L418&374   AOD increase 6.1 times, do you mean "increased by a factor of 6.1?"
**Response:** "the AOD increased 6.1 times" has been modified to "the AOD increased by a factor of 6.1".

L421   variations
**Response:** "variation" has been modified to "variations".
* * *
**References:**

Baars, H., Seifert, P., Engelmann, R., and Wandinger, U.: Target categorization of aerosol and clouds by continuous multiwavelength-polarization lidar measurements, Atmos. Meas. Tech., 10, 3175–3201, https://doi.org/10.5194/amt-10-3175-2017, 2017.

Chen, J., Li, C., Ristovski, Z., Milic, A., Gu, Y., Islam, M. S., Wang, S., Hao, J., Zhang, H., He, C., Guo, H., Fu, H., Miljevic, B., Morawska, L., Thai, P., Lam, Y., Pereira, G., Ding, A., Huang, X., and Dumka, U. C.: A review of biomass burning: Emissions and impacts on air quality, health and climate in China, Sci. Total Environ., 579, 1000–1034, https://doi.org/10.1016/j.scitotenv.2016.11.025, 2017.

Chen, Y., Chen, S., Bi, H., Zhou, J., and Zhang, Y.: Where is the Dust Source of 2023 Several Severe Dust Events in China?, Bull. Amer. Meteor. Soc., 105, E2085–E2096, https://doi.org/10.1175/BAMS-D-23-0121.1, 2024.

ECMWF: IFS Documentation CY43R3 - Part IV: Physical processes, https://doi.org/10.21957/efyk72kl, 2017.

Gui, K., Yao, W., Che, H., An, L., Zheng, Y., Li, L., Zhao, H., Zhang, L., Zhong, J., Wang, Y., and Zhang, X.: Record-breaking dust loading during two mega dust storm events over northern China in March 2021: aerosol optical and radiative properties and meteorological drivers, Atmos. Chem. Phys., 22, 7905–7932, https://doi.org/10.5194/acp-22-7905-2022, 2022.

Haarig, M., Ansmann, A., Baars, H., Jimenez, C., Veselovskii, I., Engelmann, R., and Althausen, D.: Depolarization and lidar ratios at 355, 532, and 1064 nm and microphysical properties of aged tropospheric and stratospheric Canadian wildfire smoke, Atmos. Chem. Phys., 18, 11847–11861, https://doi.org/10.5194/acp-18-11847-2018, 2018.

Jing, D., He, Y., Yin, Z., Liu, F., and Yi, F.: Long-term characteristics of dust aerosols over central China from 2010 to 2020 observed with polarization lidar. Atmos. Res., 297, 107129, https://doi.org/10.1016/j.atmosres.2023.107129, 2024.

Ministry of Ecology and Environment of the People's Republic of China: Technical guidelines for air pollution control projects, available at: https://www.mee.gov.cn/ywgz/fgbz/bz/bzwb/other/hjbhgc/201012/t20101224_199112.shtml (last access: 16 August, 2025), 2010 (in Chinese).

Seidel, D. J., Zhang, Y., Beljaars, A., Golaz, J. C., Jacobson, A. R., and Medeiros, B.: Climatology of the planetary boundary layer over the continental United States and Europe, J. Geophys. Res. Atmos., 117, https://doi.org/10.1029/2012JD018143, 2012.

Torres, O., Tanskanen, A., Veihelmann, B., Ahn, C., Braak, R., Bhartia, P. K., Veefkind, P., Levelt, P.: Aerosols and surface UV products from Ozone Monitoring Instrument observations: An overview, J. Geophys. Res. Atmos., 112, https://doi.org/10.1029/2007JD008809, 2007.

Yin, Z., Yi, F., Liu, F., He, Y., Zhang, Y., Yu, C., and Zhang, Y.: Long-term variations of aerosol optical properties over Wuhan with polarization lidar, Atmos. Environ., 259, 118508, https://doi.org/10.1016/j.atmosenv.2021.118508, 2021.

Ying, L., Han, J., Du, Y., and Shen, Z.: Forest fire characteristics in China: Spatial patterns and determinants with thresholds, For. Ecol. Manag., 424, 345–354, https://doi.org/10.1016/j.foreco.2018.05.020, 2018.

---

## Author Comment (AC2)

**Responses to RC2**

**General Remarks**

This study examines the variations of tropospheric aerosols observed by lidar in Wuhan from 2010 to 2024. It presents long-term trends across different periods and explores the respective contributions from natural and anthropogenic aerosol sources. In addition, two case studies are included to illustrate typical pollution events. Overall, while the manuscript is well written and appears to be a good fit for Atmospheric Chemistry and Physics (ACP), I have concerns regarding the scientific novelty and the structural coherence of the paper—specifically, the balance between long-term trend analysis and the inclusion of case studies. These issues should be addressed prior to possible publication. Please find my detailed comments below:

**Response:** We appreciate the reviewer's thoughtful review and constructive comments. We have added more clarification regarding the motivation of this work and highlighted the importance of the additional four years of observation in the responses, as given below (please see the third paragraph of Section 1). An analysis of meteorological and anthropogenic contributions to the AOD trend over Wuhan during the two stages was added (please see the relevant text to newly-added Figure 6 in Section 3.3) to explain the specific factors that impact the AOD variations. Relevant statements have also been revised to make the logic of the article coherent. All the comments have been addressed accordingly with the modifications in red color in the revised manuscript, and the responses to the individual comments are given in blue color as below.

**Specific comments**

**Comment:** How is DOD determined? I would like to see the seasonal variations of DOD in Wuhan. Anthropogenic activities also cause dust emissions and the assumption that DOD is totally natural should be justified.

**Response:** Thank you for pointing this out. We first divide the lidar-derived backscatter coefficients into the contributions from dust and the non-dust component. (Tesche et al., 2009). By assuming a fixed lidar ratio for dust and non-dust, their respective extinction coefficients can be obtained. Then, DOD can be calculated by integrating the lidar-derived dust extinction coefficient within 0-7 km. The Equations for calculating AOD and DOD have been added in the revised manuscript as follows "**In this study, the tropospheric AOD and dust optical depth (DOD) can be calculated by:**

$$AOD = \int_{Z_b}^{Z_t} \alpha_p(z)dz \tag{1}$$

$$DOD = \int_{Z_b}^{Z_t} \alpha_d(z)dz \tag{2}$$

where $Z_b$ and $Z_t$ are the lower (base) and upper (top) limits of the integration height, respectively. The $Z_b$ was set to 0 km, extinction coefficients below 0.35 km were assumed equal to that at 0.35 km, possibly causing an uncertainty of <0.05 in AOD (Baars et al., 2017). The $Z_t$ was set to 7 km to ensure a sufficient signal-to-noise ratio (Yin et al., 2021b). The non-dust AOD was derived by subtracting DOD from AOD.** (please see lines 92-98)

The seasonal variations of DOD in Wuhan have been analyzed in our previous study (Jing et al., 2024) as follows. "*The largest seasonal average DOD occurs in spring (0.21), followed by winter (0.15) and autumn (0.08). In summer, dust aerosols are rarely observed over Wuhan with a much smaller DOD of 0.02, attributed to the prevalence of the southeastern summer monsoon that inhibits the southeastward transport of dust particles.*" For clarity, we have also added the following sentences regarding the seasonal variations of DOD in the revised manuscript. "**Dust aerosols mainly originate from major desert regions in East Asia, i.e., the GD and TD, and generally intrude into Wuhan during spring (DOD: 0.21) and winter (DOD: 0.15). In summer, however, the prevailing southeastern monsoon suppresses the southeastward transport of dust**

**plumes, leading to extremely rare dust intrusions, with a DOD of 0.02, i.e., an order of magnitude lower than in spring (Jing et al., 2024).”** (please see lines 241-244)

[Figure]

**Fig. 6.** Box plot of monthly mean DOD in spring, summer, autumn, and winter over Wuhan from October 2010 to June 2020.

Figure 1R. Seasonal variations in DOD over Wuhan during 2010 -2020 (Jing et al., 2024).

Thank you very much for pointing out the potential role of anthropogenic dust. Indeed, our polarization lidar observations alone cannot completely exclude the possible contribution of anthropogenic dust emissions in Wuhan. As a megacity, Wuhan is mainly affected by direct anthropogenic dust from human activities, which differs from indirect anthropogenic dust originating from non-urbanized surfaces such as cropland, pastureland, and dry lakes. Chen et al. (2019) investigated global direct anthropogenic dust emissions and found that the main sources in China are concentrated in the North China Plain, Loess Plateau, and Northeast China, as seen from Figure 2R below. According to their results, we can infer that anthropogenic dust contributions over central China, where Wuhan is located, are nearly negligible. Accordingly, we have added the following statements to the revised manuscript. **“In addition, we cannot completely rule out the potential contribution of anthropogenic dust to the lidar-derived DOD. In a megacity such as Wuhan, direct anthropogenic dust is likely more important than indirect anthropogenic dust emitted from non-urbanized surfaces, e.g., cropland, pastureland, and dry lakes. However, Chen et al. (2019) reported that the major direct anthropogenic dust emissions in China are concentrated in Northern China, while emissions over central China are significantly lower. Therefore, it is reasonable to assume that the lidar-derived DOD primarily reflects contributions from desert dust (i.e., natural sources).”** (please see lines 244-249)

[Figure]

Figure 2R. The spatial distributions of potential direct anthropogenic dust sources at the global scale from 2007 to 2010 (Chen et al., 2019).

**Comment:** The authors have published a similar paper on 2010-2020 AOD variations over Wuhan and this work is just an extension from their previous study. As such, the scientific motivation of this current analysis is not very clear.

**Response:** In our previous study, we reported a consistent downward trend in AOD during 2010–2020. However, it can be observed that this AOD decline appeared to cease and even slightly reverse since 2018 (Figure 3R, Yin et al., 2021). Given that the reduction in surface $PM_{2.5}$ concentrations after 2018 has slowed (Geng et al., 2024), it is difficult to conclude whether the consistent downward trend in AOD merely slowed or completely halted at some point after 2018. In addition, in the additional four years following 2020, several factors, including industrial shutdowns due to the COVID-19 pandemic, abnormal Asian dust events, and extreme precipitation, may have influenced AOD levels in Wuhan. Only with these four additional years of lidar observations can we clearly identify a fluctuation period from 2018 to 2024, and further conduct an analysis that can distinguish the respective contributions of meteorological and anthropogenic factors (as presented in the response to the next comment). This presents an important finding that distinguishes the current work from our previous study. For clarity, we have emphasized the abovementioned motivation in the Introduction Section of the revised manuscript. (please see lines 46-53)

[Figure]

Figure 3R. Variations in 532-nm AOD derived during 2010-2020 (Yin et al., 2021).

Moreover, analyses in this study are based on our first-hand, home-made lidar measurements collected by our team consistently over the past 15 years. As shown in Figure 1, from October 2010 to September 2024, there were 2825 observation days, and a total of 24910 valid cloud-free profiles were obtained from 2139 days, which is 91% and 85% more than 1478 observation days and 1159 days with valid profiles in Yin et al. (2021). In recent years, data coverage has increased substantially, with more than 300 observation days annually (including complete observation records during the initial stage of the COVID-19 lockdown in Wuhan city) and over 250 days yielding valid retrieved profiles of aerosol optical properties. These additional four years of observations provide an important extension to the long-term record of height-resolved aerosol optical properties over central China.

[Figure]

Figure 4R. Annual count of days with lidar observations and valid retrievals of aerosol optical parameter profiles (October 2010 to September 2024).

**Comment:** More explanations on the AOD trends are needed. Why there was a fluctuating trend in stage II? How to isolate the role of meteorological influence? The driving factors were not well elucidated.

**Response:** Thank you very much for this constructive comment. We have discussed the DOD variation trend in Section 3.2, which reflects climate change in desert areas. For non-dust AOD variation, we have now added an analysis of the contributions from the meteorological and anthropogenic factors in Section 3.3, applying the Lindeman, Merenda, and Gold (LMG) method from Che et al. (2019). The relative contributions of meteorological or anthropogenic-emission factors to non-dust AOD variation, along with the associated partial correlation analysis, are presented in the newly-added Figure 6 of the revised manuscript. The primary meteorological factors considered include ERA5 hourly relative humidity (%), u- and v-component wind speeds (WS, m s$^{-1}$) within 1000-850 hPa, total precipitation (Pre, m), vertical velocity (VV, Pa s$^{-1}$) at 850 hPa, and boundary layer height (BLH, m). Surface-measured PM$_{2.5}$ concentrations were used to reflect the contribution of anthropogenic emissions to non-dust AOD variation. Detailed methodological descriptions have been provided in Section 2.6 of the revised manuscript (please see lines 170-180), and the corresponding analyses have been added to Section 3.3 (please see lines 316-333)

[Figure]

Figure 5R. (Figure 6 in revised manuscript) The LMG method-estimated relative contributions (%) of monthly mean meteorological variations and PM2.5 concentration on monthly mean non-dust AOD during (a) 2014-

2017 and (c) 2018-2024. The partial correlation analyses between non-dust AOD and variations during 2014-2017 and 2018-2024 in Figure 6a (or c) were presented by Figure 6 b and d, respectively.

**Comment:** In addition, I can't find the linkage between trend analysis and the two case studies. Was aerosol pollution in June 2014 and January 2019 the worst ones? Similar pollution events have been well studied.

**Response:** These two cases represent typical summer and winter air pollution episodes over central China, frequently observed during our long-term measurements. They correspond to the seasonal variation of anthropogenic aerosols discussed in Section 3.3: enhanced particle extinction coefficient at altitudes of 0.7-2.5 km in summer and below 0.7 km in winter, reflecting transboundary aerosol intrusion and local pollutions, respectively.

The summertime transboundary agricultural biomass burning smoke (ABBS) case in June 2014 shows how smoke from straw burning was transported to Wuhan, resulting in severe air pollution. This event, along with another similar ABBS intrusion event in June 2012 (Zhang et al., 2014), helps explain the enhanced AOD values observed in early summer of the year. Following the implementation of straw-burning bans by the Chinese government, no extreme events with AOD >0.8 have been recorded after 2015.

By presenting the two cases (i.e., one transboundary-driven and the other locally driven), we clearly distinguish and quantify the differential impacts of regional transport and local emissions on AOD under varying conditions. While long-term statistics capture the overall trends, these case studies better reveal the dominant mechanisms behind them. To better link the trend analysis with the two case studies, we have revised the related statements as follows. "**As discussed in Section 3.3, the seasonal characteristics of anthropogenic aerosols show enhanced extinction coefficients at higher altitudes in summer and near the surface in winter. This reflects two typical aerosol pollution patterns: transboundary aerosol intrusion in summer and local pollution in winter**" (please see lines 369-371) and "**Here, we present two typical air pollution cases: summertime transboundary ABBS in June 2014 and wintertime local anthropogenic aerosol pollution in January 2019. These cases provide valuable insights into the dominant mechanisms driving the long-term AOD trends.**" (please see lines 373-375)
* * *
**References:**

Che, H., Gui, K., Xia, X., Wang, Y., Holben, B. N., Goloub, P., Cuevas-Agulló, E., Wang, H., Zheng, Y., Zhao, H., and Zhang, X.: Large contribution of meteorological factors to inter-decadal changes in regional aerosol optical depth, Atmos. Chem. Phys., 19, 10497–10523, https://doi.org/10.5194/acp-19-10497-2019, 2019.

Chen, S., Jiang, N., Huang, J., Zang, Z., Guan, X., Ma, X., Luo, Y., Li, J., Zhang, X., and Zhang, Y.: Estimations of indirect and direct anthropogenic dust emission at the global scale, Atmos. Environ., 200, 50–60, https://doi.org/10.1016/j.atmosenv.2018.11.063, 2019.

Geng, G., Liu, Y., Liu, Y., Liu, S., Cheng, J., Yan, L., Wu, N., Hu, H., Tong, D., Zheng, B., Yin, Z., He, K., and Zhang, Q.: Efficacy of China's clean air actions to tackle $PM_{2.5}$ pollution between 2013 and 2020, Nat. Geosci., 17, 987–994, https://doi.org/10.1038/s41561-024-01540-z, 2024.

Jing, D., He, Y., Yin, Z., Liu, F., and Yi, F.: Long-term characteristics of dust aerosols over central China from 2010 to 2020 observed with polarization lidar. Atmos. Res., 297, 107129, https://doi.org/10.1016/j.atmosres.2023.107129, 2024.

Yin, Z., Yi, F., Liu, F., He, Y., Zhang, Y., Yu, C., and Zhang, Y.: Long-term variations of aerosol optical properties over Wuhan with polarization lidar, Atmos. Environ., 259, 118508, https://doi.org/10.1016/j.atmosenv.2021.118508, 2021.

Zhang, M., Ma, Y., Gong, W., and Zhu, Z: Aerosol optical properties of a haze episode in Wuhan based on ground-based and satellite observations, Atmosphere 5, 699–719, https://doi.org/10.3390/atmos5040699, 2014.

---

## Author Response (AR2)

**Responses to RC1**

**General Remarks**

I would like to suggest this manuscript to be published as ACP Measurement report.

Response: Thank you very much for the reviewer's suggestion. From our perspective, this manuscript is still better suited as a research article for ACP rather than a measurement report. In Yin et al. (2021), we previously reported a consistent downward trend in AOD during 2010–2020. However, an open question remains as to whether this downward trend slowed or even halted after 2018, especially considering the slowed reduction in surface PM2.5 since 2018 (Geng et al., 2024). The four additional years of lidar observations included in this study enable us to clearly identify a fluctuation period (of AOD) from 2018 to 2024. In addition, we also separate the contributions from dust (natural) and non-dust (anthropogenic) components and analyze their respective evolution during the study period. Importantly, the analyses presented here are based on first-hand, home-made lidar measurements collected consistently by our team over the past 15 years. These fixed-located, high-resolution, and height-resolved aerosol optical properties over central China provide an important basis for evaluating the effectiveness of emission control policies. Finally, this study includes extensive physical analyses for the contributions of meteorological and anthropogenic factors to variations in non-dust (anthropogenic) AOD.

Therefore, while we highly respect and appreciate the reviewer's suggestions, we believe that our manuscript meets the criteria for a research article. We respectfully leave the final decision to the editor.

**References:**

- Geng, G., Liu, Y., Liu, Y., Liu, S., Cheng, J., Yan, L., Wu, N., Hu, H., Tong, D., Zheng, B., Yin, Z., He, K., and Zhang, Q.: Efficacy of China's clean air actions to tackle PM2.5 pollution between 2013 and 2020, Nat. Geosci., 17, 987–994, https://doi.org/10.1038/s41561-024-01540-z, 2024.
- Yin, Z., Yi, F., Liu, F., He, Y., Zhang, Y., Yu, C., and Zhang, Y.: Long-term variations of aerosol optical properties over Wuhan with polarization lidar, Atmos. Environ., 259, 118508, https://doi.org/10.1016/j.atmosenv.2021.118508, 2021.

**Responses to RC2**

**General Remarks**

The revised paper has been significantly improved. The authors responded to comments from two reviewers (RC1 and RC2) regarding the manuscript on the evolution of aerosols over Wuhan (2010-2024): To RC1, they clarified the basis for dividing AOD (Stage I: 2010-2017, declining; Stage II: 2018-2024, fluctuating) and DOD (downtrend extended to 2020) stages, improved the clarity of case studies by adjusting figure order, adding HYSPLIT backward trajectories, and supplementing comparative explanations of satellite and lidar data, while revising text wording, figure captions, and data processing descriptions. To RC2, they supplemented the calculation method and seasonal variations of DOD, justified that DOD is mainly from natural dust, emphasized the value of 4 additional years of observation data (2021-2024) to highlight innovation, added analysis of meteorological and anthropogenic contributions to AOD trends using the LMG method, and linked the two case studies (transboundary agricultural biomass burning smoke in June 2014, local haze in January 2019) to long-term trend analysis, ultimately improving the manuscript through text revisions and reference supplements. However, there are still some issues that require improvement.

**Response:** We appreciate the reviewer's thoughtful review of our first-round responses and revised manuscript. We have responded to the remaining two comments (in blue as below) and revised the manuscript accordingly (in red in the revised manuscript).

**Specific comments**

Comment: Page 10, line 250-254, and Page 21, line 472-475. The authors note that DOD declined until 2020 (-0.011 yr-1) and rebounded after 2021, while AOD entered a fluctuating phase as early as 2018. However, the contribution of post-2020 DOD rebound to AOD fluctuations was not quantified. For example: What percentage of the monthly AOD increase in the extreme spring dust events of 2021 was attributed to DOD increments? What is the explanatory power of interannual DOD changes (2021–2024) for total AOD fluctuations? Without this analysis, the relative importance of natural (dust) and anthropogenic (non-dust) sources in driving Stage II AOD fluctuations cannot be clarified, leading to an underdeveloped conclusion on "differences in driving factors between stages."

**Response:** Thank you very much for the valuable comments. Regarding lines 472-475, we only discuss the evolution of non-dust AOD, which does not include the influence of dust (as stated in the first sentence of that paragraph). In this study, we first present the evolution of total AOD (Figure 2b), and then separate it into the respective evolutions of DOD (dust component; Figure 3) and non-dust AOD (non-dust component; Figure 4a). From our perspective, this storyline clearly illustrates the relative importance of natural (dust) and anthropogenic (non-dust) sources in driving Stage II AOD fluctuations.

In addition, we agree with the reviewer that, during 2021-2024 when DOD increased and non-dust AOD fluctuated, it is reasonable to discuss the potential contribution of the post-2020 DOD rebound to the total AOD fluctuations. Therefore, in the revised manuscript, we have added the ratio of DOD to total AOD in the new Figure 3b (Figure 1R b here), along with the following statements "The ratio of DOD to total AOD is presented in Figure 3b. The monthly mean DOD fraction can reach up to 60% during dust-intrusion seasons in spring and winter (e.g., in 2012, 2018, and 2021). In contrast to the pronounced trend in DOD values, the annual mean fraction of DOD fluctuated only between approximately 20% to 30% over the entire period, indicating that the contribution of dust to total AOD remained relatively stable despite year-to-year variations." (please see lines 268-271).

Figure 1R. Monthly mean (a) DOD and (b) the ratio of DOD to total AOD from 2010 to 2024. A box plot is presented for each year during 2011-2024, with the gray solid dots representing the annual mean values. The dark orange line represents the linear fit of the monthly mean DOD from 2010 to 2020.

Comment: Page 21, line 472-481. The authors generally conclude that "Stage II AOD fluctuations are driven by meteorological and anthropogenic factors" but did not disaggregate dominant factors by year. For example: Was the 2019 AOD increase mainly due to reduced precipitation (weakened wet scavenging)? Was the 2021 spring AOD rise dominated by dust increments? Did 2023 AOD fluctuations relate to regional transport (e.g., resurgent biomass burning)? Without year-specific driver analysis, conclusions are overly broad and fail to provide targeted support for "differentiated pollution control strategies".

**Response:** The statsments in lines 472-481 refer to the non-dust components; therefore, the dust intrusion during 2021 spring may not be included in the discussions here. To address the reviewer's concern, we have added the annual variations in the contributions of meteorological and anthropogenic factors to non-dust AOD throuth the LMG method (Figure 2R below and newly-added Figure 7 in the revised manuscript). Note than after 2022, RH has the dominant contribution. The meteorological influences in specific years are now discussed accordingly.

The following paragraph has been added in the revised manuscript "In addition, the annual variation in the LMG method-derived relative contributions of daily mean meteorological and authropogenic factors are presented in Figure 7. The trends of the dominant factors, RH and PM2.5, show increasing and decreasing patterns from Stage I to Stage II, respectively. Overall, the contribution of RH is larger than that of PM2.5 except in 2015. In Stage II, the notable increase in non-dust AOD in 2018 is associated with the increased contribution of PM2.5, which rises to 21% compared with 8% in 2017. In 2019, the contribution of vertical velocity is comparable to that of PM2.5, which is inferred to be related to regional pollution transportation. From 2020 to 2021, the contribution of PM2.5 decreases due to the COVID-19 lockdown. In 2021, wind speed contributes more than PM2.5, likely due to enhanced air cleansing by cold air transported from high latitudes under the 'Warm Arctic-Cold Siberia' pattern (Zhang et al., 2021b). After 2022, the contribution of PM2.5 increases to around 20%, coinciding with the resumption of industrial activities and production. It is worth noting that the contribution of RH is around 60% in Stage

II, even exceeds 70% in 2023, which is higher than the value shown in Figure 6c. This difference arises because the contributions in Figure 7 are based on daily mean values rather than monthly means. Under these circumstances, the daily variability of RH is more pronounced than that of other meteorological variables, and the influence of hygroscopic growth on aerosol backscatter and thus AOD is more direct." (please see lines 343-354).

Figure 2R. Annual variations in the LMG method-estimated relative contributions (%) of daily mean meteorological variations and PM2.5 concentration on daily mean non-dust AOD.